



# Contributions of the direct supply of belowground seagrass detritus and trapping of suspended organic matter to the sedimentary organic carbon stock in seagrass meadows

Toko Tanaya[1], Kenta Watanabe[1], Shoji Yamamoto[2], Chuki Hongo[3], Hajime Kayanne[2], Tomohiro Kuwae[1]

[1]Coastal and Estuarine Environment Research Group, Port and Airport Research Institute, 3-1-1 Nagase, Yokosuka 239-0826, Japan
[2]Department of Earth and Planetary Science, the University of Tokyo, Hongo 7-3-1, Bunkyo-ku, Tokyo 113-0033, Japan
[3]Department of Chemistry, Biology, and Marine Science, University of the Ryukyus, Senbaru 1, Nishihara, Okinawa 903-
10   0213, Japan

*Correspondence to*: Toko Tanaya (tanaya-t@ipc.pari.go.jp)





**Abstract.** Carbon captured by marine living organisms is called "blue carbon", and seagrass meadows are a dominant blue carbon sink. However, our knowledge of how seagrass increases sedimentary organic carbon (OC) stocks is limited. We investigated two pathways of OC enrichment: trapping of organic matter in the water column and the direct supply of belowground seagrass detritus. We developed a new type of box corer to facilitate the retrieval of intact cores that preserve the

structures of both sediments (including coarse sediments and dead plant structures) and live seagrass bodies. We measured seagrass density, total OC mass ($OC_{total}$) [= live seagrass OC biomass ($OC_{bio}$) + sedimentary OC mass ($OC_{sed}$)], and the stable carbon isotope ratio ($\delta^{13}C$) of $OC_{sed}$ at back-reef and estuarine sites in the tropical Indo-Pacific region. $OC_{bio}$ accounted for 19% and $OC_{sed}$ for 81% of $OC_{total}$; this contribution of $OC_{bio}$ to $OC_{total}$ is the highest in globally compiled data. Belowground detritus accounted for ~90% of the OC mass of dead plant structures (>2 mm in size) ($OC_{dead}$). At the back-reef site,

belowground seagrass biomass, $OC_{dead}$, and $\delta^{13}C$ of $OC_{sed}$ ($\delta^{13}C_{sed}$) were positively correlated with $OC_{sed}$, indicating that the direct supply of belowground seagrass detritus is a major mechanism of $OC_{sed}$ enrichment. At the estuarine site, aboveground seagrass biomass was positively correlated with $OC_{sed}$ but $\delta^{13}C_{sed}$ did not correlate with $OC_{sed}$, indicating that trapping of suspended OC by seagrass leaves is a major mechanism of $OC_{sed}$ enrichment there. We inferred that the relative importance of these two pathways may depend on the supply (productivity) of belowground biomass. Our results indicate that belowground

biomass productivity of seagrass meadows, in addition to their aboveground morphological complexity, is an important factor controlling their OC stock. Consideration of this factor will improve global blue-carbon estimates.



## 1 Introduction

The carbon captured by marine living organisms has been termed "blue carbon" (Nelleman et al., 2009). Among marine ecosystems, the organic carbon (OC) accumulation rate of vegetated coastal systems such as seagrass meadows, mangrove forests, and salt marshes is estimated to be higher than that of terrestrial forests (Mcleod et al., 2011). The global total OC stock contained in the top 1 m of sediment and in the plant biomass in these vegetated ecosystems is estimated to be 0.63–8.54 Pg C (Pendleton et al., 2012). Thus, vegetated ecosystems are expected to contribute greatly to the mitigation of global warming. In this regard, seagrass meadows have attracted particular attention because they are one of the most dominant blue-carbon sinks (Kennedy et al., 2010; Fourqurean et al., 2012). However, the OC stock of a seagrass meadow is highly variable, depending on geographical region (Miyajima et al., 2015), seagrass species (Lavery et al., 2013), microlocation within a seagrass patch (Ricart et al., 2015), and the patch scale (Miyajima et al., 2017). Hence, to develop a precise methodology of OC estimation and reduce the uncertainty of the global estimate, it is necessary to understand the factors controlling OC stocks in seagrass meadows (Duarte et al., 2013).

Seagrass meadows enhance the accumulation of sedimentary OC by directly supplying of abundant OC from their high production (Duarte et al., 2010), by reducing sediment resuspension, and by promoting sedimentation of autochthonous and allochthonous OC in the water column (Agawin and Duarte, 2002; Gacia and Duarte, 2001; Gacia et al., 2003; Hendriks et al., 2008). However, our knowledge of the factors that mediate the sequestration of sedimentary OC by seagrass meadows is limited. For example, the chemical recalcitrance of the supplied organic matter (Watanabe and Kuwae, 2015) and the specific surface area of the sediment (Miyajima et al., 2017) are factors that control the sedimentary OC stock in seagrass meadows. Recent studies have also shown that, in addition to chemical and physical factors, biological factors such as primary productivity, seagrass shoot density, and the amount of leaf material (as indicated by the leaf area index) also affect the sedimentary OC stock (Samper-Villarreal et al., 2016; Serrano et al., 2014; Serrano et al., 2016b). In addition, an increase in the amount of leaf material may enhance the trapping of suspended OC and, thus, the accumulation of sedimentary OC (Dahl et al., 2016; Gacia et al., 1999). An increase in seagrass density may also cause an increase in seagrass production per unit area and thus enhance the direct supply of seagrass-derived OC. However, few previous studies have analyzed the controlling factors and provenance of sedimentary OC along a seagrass biomass gradient (Kennedy et al., 2004; Kennedy et al., 2010; Samper-Villarreal et al., 2016; Howard et al., 2017). Kennedy et al. (2004, 2010) and Howard et al. (2017) found no significant relationship between seagrass biomass and sedimentary OC, whereas Samper-Villarreal et al. (2016) concluded that autochthonous sedimentary OC increased as the leaf area index increased. However, they did not show the mechanism (pathway) by which seagrass-derived OC became sedimentary OC; that is, they did not show whether the seagrass trapped seagrass-derived OC suspended in the water column or directly supplied seagrass-derived carbon to the sediments.

To assess the effect of seagrass on the sedimentary OC stock, it is important to examine all stock components, including live and dead above- and belowground biomass in the sediment column, and their origins. For this reason, it is necessary to retrieve intact cores, because both macroscopic plant materials (Miyajima et al., 1998) and OC derived from



calcareous organisms such as corals, foraminifera, molluscs, and coralline algae (Ingalls et al., 2003; Versteegh et al., 2011) occur in the coarse sediment fraction (sand and gravels), especially in tropical seagrass meadows around coral reefs (Suzuki, 2005). However, to our knowledge, all previous studies have only examined some of the stock components: for example, the fine sediment fraction (<1–2 mm diameter) (Hemminga et al., 1994; Miyajima et al., 2015; Kennedy et al., 2004; Ricart et al.,

5    2015), dead plant structures (Cebrian et al., 2000), surface sediment (Barron et al., 2004), and small subsamples from a core (Dahl et al., 2016).

In this study, to investigate the relationship between seagrass and the sedimentary OC stock, we used intact cores that included all seagrass live and dead bodies and sediments and then performed the OC mass and stable carbon isotope analyses of all components of the cores to examine the origin of the OC.

## 2 Materials and Methods

### 2.1 Study sites

To assess the relationship between seagrass and the sedimentary OC stock, we chose tropical Indo-Pacific seagrass meadow sites. Globally, the tropical Indo-Pacific region is the world's largest bioregion and contains the highest diversity of seagrasses,

which are distributed predominantly on coral reef flats (Short et al., 2007). Globally, the total documented seagrass area is 164,000 km$^2$ (Green and Short, 2003), and the total seagrass area in the Indo-Pacific region, excluding Australia, where both tropical and temperate seagrasses are distributed, is around 32,400 km$^2$, or about 20% of the total area. Furthermore, given that about half of the documented seagrass habitat in Australia is composed of tropical seagrasses (Kirkman, 1997), the total area of tropical Indo-Pacific seagrass habitat reaches approximately 116,000 km$^2$, accounting for 70% of the global seagrass area.

Thus, accurate estimation of the blue-carbon stock of seagrasses in the tropical Indo-Pacific region is important for the estimation of the global seagrass carbon stock. However, in spite of the geographical importance of this region, reports on seagrass OC stocks there are limited (Lavery et al., 2013; Miyajima et al., 2015).

We obtained cores from two Indo-Pacific tropical seagrass meadow sites from 13 to 23 August 2014. The two sites, a back-reef site (Shiraho reef) and an estuarine site (Fukido estuary), both located around Ishigaki Island, Okinawa,

southwestern Japan, (Fig. 1), have different allochthonous carbon input amounts. The back-reef site is situated on a well-developed reef flat about 1 km wide, where seagrass meadows, dominated by *Thalassia hemprichii*, are distributed between 100 and 300 m from the shoreline. The site is about 2 km south of the mouth of the Todoroki River, and most sediments transported by the river accumulate on its north side (Mitsumoto et al., 2000) because the prevailing current, which is controlled by large channels in the reef, is northward (Tamura et al., 2007). Therefore, terrestrial sediment input to the back-reef site is

low. The mud (silt + clay) content of the surface sediment of the seagrass meadows at the site ranges from 1.2% to 3.9% (mean 2.3%) (Tanaka and Kayanne, 2007). The estuarine site is located near the mouth of a small river, which is bordered by small



mangrove forests. The freshwater inflow is low, so water exchange between the river and estuary is controlled mainly by tidal motion (Terada et al., 2007). The dominant seagrass species at the site is *Enhalus acoroides*. The mud content of the surface sediment in the seagrass meadows at the estuarine site ranges from 0.9% to 6.4% (mean 3.6%) (Tanaka and Kayanne, 2007).

**2.2 Core sampling**

We developed a new box corer to facilitate the retrieval of intact cores that preserve sedimentary structures as well as above- and belowground live and dead seagrass bodies (Fig. 2). The box corer is 15 cm wide, 15 cm deep, and 17 cm high and is made of stainless steel so that it can cut through roots and rhizomes. A shutter 1 cm above the bottom of the corer is designed to cut through the relatively hard belowground seagrass bodies, making it possible to obtain intact cores. The corer also has a

10 lid to prevent the loss of surficial sediments from the core during underwater sampling. The corer is large enough to retrieve all components of the OC stock whole: shoots, live and dead above- and belowground seagrass bodies, and old skeletal OC in sand and gravel derived from calcareous organisms such as corals, foraminifera, molluscs, and coralline algae. Most cores obtained with the corer were about 15 cm long, but we were not able to insert the core to its full length at three sampling points because of the presence of large gravels in the sediment. We were able to collect all of the seagrass biomass at these points,

however.

To measure the total OC mass ($OC_{total}$), we quantified three components of the box corer samples (Fig. 3): (1) live seagrass bodies ($OC_{bio}$); (2) dead plant structures (>2 mm in size: dead seagrass leaves, sheaths, rhizomes, and roots detached from live structures) ($OC_{dead}$); and (3) coarse (>1 mm diameter) sediments (excluding dead plant structures >2 mm in size) ($OC_{csed}$). We also collected samples with cylinder cores so that we could obtain depth profiles of OC in the fine (<1 mm

diameter) sediments (including dead plant structures <1 mm in size) ($OC_{fsed}$). It was technically impossible to obtain these profiles with the box corer because of its large surface sampling area and the high density of the belowground structures (Fig. 2c). The samples retrieved by the box corer were immediately sieved through a 1mm mesh sieve *in situ* to obtain the >1 mm fractions of the $OC_{bio}$, $OC_{dead}$ and $OC_{csed}$. Seagrass bodies have air-filled lacunae so that they float; thus, we considered buoyant seagrass bodies captured by the sieve to be $OC_{bio}$ (Borum et al., 2006). We merged any dead plant structures attached to live

seagrass bodies into $OC_{bio}$ because their mass was usually very small. We collected a cylinder core 10–16 cm long with an acrylic pipe (internal diameter 6.6 cm) from a point immediately adjacent to each box core. We subdivided each cylinder core into 1-cm-long subsamples from the surface to the bottom of the core.

We obtained 20 paired samples (one box and one cylinder core) from the back-reef site and eight paired samples from the estuarine site. At the back-reef site, we collected 16 paired samples from vegetated points in the seagrass meadows, two

from bare patches in the seagrass meadows, and two from unvegetated areas (Fig. 1b). Similarly, at the estuarine site, we collected five paired samples from vegetated points, one pair from a bare point, and two paired samples from unvegetated areas near the river mouth (Fig. 1c).



Potential sources of sedimentary OC (OC$_{sed}$) were also collected at both sites and analyzed for δ$^{13}$C. Samples of seagrass leaves were collected from all dominant seagrass species at each site: *T. hemprichii*, *Cymodocea rotundata*, *C. serrulata*, and *Halodule uninervis* at the back-reef site, and *E. acoroides*, *T. hemprichii*, and *C. serrulata* at the estuarine site. Samples for determining the δ$^{13}$C of algae and coral were taken from epiphytes, benthic microalgae, and the dominant coral

species (mainly *Acropora* spp. and *Porites* spp.) at the sites. Epiphytes were collected from the seagrass leaves by using a stainless steel scraper, and benthic microalgae were extracted from the surface sediment (up to approximately 1-mm depth) by the method of Kuwae et al. (2008). All obtained samples were stored in polyethylene bags at −20 °C until analysis.

We used the published δ$^{13}$C data of suspended OC (collected about 1 km off the outer reef edge of Ishigaki Island) and of terrestrial particulate organic matter (POM; collected from the Fukido River, Ishigaki Island) from Miyajima et al.

(2015). We assumed that the published δ$^{13}$C data were normally distributed.

**2.3 OC and stable isotope analysis**

We identified live seagrass bodies to the species level and separated aboveground biomass (leaf blades) from belowground biomass (leaf sheathes, rhizomes, and roots). Then we dried all parts at 60 °C and weighed them. Box corer sediments were

15 dried at 60 °C and sieved through a 2-mm-mesh sieve, and the included dead plant structures (>2 mm in size) were picked out and weighed. To ensure homogeneity of subsamples, the coarse sediments (excluding the dead plant structures) were first crushed to approximately 1-mm grains with a jaw crusher (Jaw Crusher PULVERISETTE 1 Model I classic line, FRITSCH, Ltd., Idar-Oberstein, Germany) and then divided into 16 or 64 subsamples with a splitter (Simple microsplitter, Iwamoto Mineral, Ltd., Tokyo, Japan). The cylinder core samples were subdivided into surface (0−1 cm depth), intermediate (5−8 cm

depth), and bottom (9−16 cm depth) layers and dried at 60 °C. For the OC$_{fsed}$ analysis, each layer was sieved through a 1mm mesh sieve and then subdivided into two or four subsamples with the splitter. All subsamples used for chemical analyses were weighed and then powdered and homogenized in an agate mill.

For OC analysis, the homogenized samples were placed in silver containers (to prevent the loss of acid-soluble OC in carbonate sediments) and pretreated with hydrochloric acid to remove carbonates (Yamamuro and Kayanne, 1995). First,

each sample was weighed in a silver container and its weight was adjusted to about 20 mg. Then, 1 N HCl was carefully and gradually added until bubbles were no longer seen, and the sample was dried at 60 °C overnight and at 105 °C for 1 h. The dried sample was then wrapped in tin foil. We measured the total OC concentration and the stable carbon isotopic ratio of each sample with an elemental analyzer-connected isotope ratio mass spectrometer (FLASH EA 1112 / DELTA$^{plus}$ Advantage, Thermo Electron, Inc., Massachusetts, USA). The stable carbon isotope ratio (δ$^{13}$C) is reported as the relative per mil deviation

from VPDB (Vienna Pee Dee Belemnite). The analytical precision of the isotope ratio mass spectrometer, based on the standard deviation of δ$^{13}$C values of internal reference replicates, was <0.2‰.



### 2.4 Determination of the mass and δ¹³C of OC

We calculated $OC_{total}$ per unit area (g C m$^{-2}$) at each sampling point by summing the $OC_{bio}$ and $OC_{sed}$ components in the top 0.15 m (Fig. 3) as follows:

$$OC_{total} = OC_{bio} + OC_{sed} . \tag{1}$$

$OC_{bio}$ was calculated as,

$$OC_{bio} = \sum_i (a_i x_i + b_i y_i) , \tag{2}$$

where $a_i$ and $b_i$ are the averaged OC concentrations (g C g$^{-1}$ DW) of the aboveground and belowground biomass, respectively, of the $i$th seagrass species collected at three different sampling points (except *C. serrulata*, which was collected at only one sampling point at the estuarine site), and $x_i$ and $y_i$ are the aboveground and belowground biomass (g m$^{-2}$), respectively, of the $i$th seagrass species. The biomasses of *Syringodium isoetifolium*, *Halophila ovalis*, and an unidentified species at the back-reef

15 site accounted for <0.1% of the total biomass, so they were excluded from this calculation. The averaged OC concentrations and aboveground and belowground biomass dry weights are summarized in Table 1.

$OC_{sed}$ was calculated as follows:

$$OC_{sed} = OC_{dead} + OC_{csed} + OC_{fsed} . \tag{3}$$

The terms of Eq. (3) were calculated by the following equations:

$$OC_{dead} = \frac{1}{100} (\%OC_{leaf} \times \rho_{leaf} + \%OC_{shrh} \times \rho_{shrh} + \%OC_{root} \times \rho_{root}) \times h , \tag{4}$$

$$OC_{csed} = \frac{1}{100} (\%OC_{csed} \times \rho_{csed}) \times h , \tag{5}$$

25 $$OC_{fsed} = \frac{1}{3} \times \frac{1}{100} (\%OC_{fseds} \times \rho_{fseds} + \%OC_{fsedm} \times \rho_{fsedm} + \%OC_{fsedb} \times \rho_{fsedb}) \times h , \tag{6}$$

where %OC is the concentration of OC (%DW) ($n = 3$); $\rho$ is the dry density (g DW m$^{-3}$) of each component (indicated by subscripts: leaf, dead leaf; shrh, dead sheath and rhizome; root, dead root; csed, coarse sediment; fseds, fine sediment of the surface layer; fsedm, fine sediment of the intermediate layer; fsedb, fine sediment of the bottom layer), and $h$ is the sample

30 thickness (0.15 m).

δ¹³C of $OC_{sed}$ (δ¹³C$_{sed}$) at each sampling point was calculated as follows:



$$\delta^{13}C_{sed} = \frac{1}{OC_{sed}} (OC_{dead} \times \delta^{13}C_{dead} + OC_{csed} \times \delta^{13}C_{csed} + \delta^{13}C_{fsed}) , \qquad (7)$$

where $\delta^{13}C_{dead}$ is the averaged $\delta^{13}C$ value of dead plant structures (sheath and rhizomes, and roots) at the back-reef and estuarine sites. We used the $\delta^{13}C_{dead}$ value at each site for the calculation of $\delta^{13}C_{sed}$. The standard deviation (SD) of $\delta^{13}C_{sed}$ derived from
the SD of $\delta^{13}C_{dead}$ was smaller than 0.1‰. $\delta^{13}C_{csed}$ is the $\delta^{13}C$ value of $OC_{csed}$. $\delta^{13}C_{fsed}$ is the averaged $\delta^{13}C$ value of $OC_{fsed}$ multiplied by the OC mass of each layer and was calculated as follows:

$$\delta^{13}C_{fsed} = \frac{1}{3} \times \frac{1}{100} (\%OC_{fseds} \times \rho_{fseds} \times \delta^{13}C_{fseds} + \%OC_{fsedm} \times \rho_{fsedm} \times \delta^{13}C_{fsedm} + \%OC_{fsedb} \times \rho_{fsedb} \times \delta^{13}C_{fsedb}) \times h . \qquad (8)$$

The averaged values of the organic carbon concentration, $\delta^{13}C$, and dry density of sediment and dead plant structures are summarized in Table 2.

## 3 Results

### 3.1 Seagrass biomass and species composition at each site

At the back-reef site, the average (±SD) aboveground and belowground biomass values were $74 \pm 45$ g DW m$^{-2}$ ($n = 16$) and $675 \pm 450$ g DW m$^{-2}$ ($n = 16$), respectively (Table 1). The dominant species was *T. hemprichii*, accounting for 76.7% of the total biomass; *C. rotundata* (18.0%), *C. serrulata* (3.3%), *H. uninervis* (1.7%), *H. ovalis* (<0.1 %), *S. isoetifolium* (<0.1%), and an unidentified species (<0.1%) were minor components at the back-reef site. At the estuarine site, the average aboveground and belowground biomass were $70 \pm 34$ g DW m$^{-2}$ ($n = 5$) and $1354 \pm 847$ g DW m$^{-2}$ ($n = 5$), respectively (Table
1). The dominant species was *E. acoroides*, accounting for 92.3% of the total biomass; *T. hemprichii* (7.0%), *C. serrulata* (0.6%), and *H. uninervis* (<0.1 %) were minor components.

### 3.2 OC density in the fine sediments

The average OC density (g C cm$^{-3}$) did not differ significantly among the fine sediment layers at either the back-reef (paired
*t*- test, Bonferroni adjusted $P > 0.05$) or the estuarine site (Wilcoxon signed rank test, Bonferroni adjusted $P > 0.05$).




### 3.3 OC mass

The averaged $OC_{bio}$, $OC_{dead}$, $OC_{fsed}$, $OC_{sed}$ and $OC_{total}$ values did not significantly differ between the sites ($OC_{bio}$, $W = 73$, $P > 0.05$; $OC_{dead}$, $W = 65$, $P > 0.05$; $OC_{fsed}$, $t = 0.67$, d.f. = 26, $P > 0.05$; $OC_{sed}$, $t = -0.52$, d.f. = 26, $P > 0.05$; $OC_{total}$, $t = -0.86$, d.f. = 26, $P > 0.05$), whereas the average $OC_{csed}$ was significantly higher at the estuarine site ($191 \pm 75$ g C m$^{-2}$) than at the back-

reef site ($123 \pm 45$ g C m$^{-2}$) ($t = -2.98$, d.f. = 26, $P = 0.006$) (Fig. 4). This higher $OC_{csed}$ at the estuarine site was resulting from the higher density of coarse sediments there than at the back-reef site ($t = -2.92$, d.f. = 26, $P = 0.007$), because the %OC of $OC_{csed}$ was not different between the sites ($W = 103$, $P > 0.05$) (Table 2). $OC_{total}$ ranged from 334 to 1785 g C m$^{-2}$ across both sites. $OC_{sed}$, which ranged from 334 to 1147 g C m$^{-2}$ and was the main component of $OC_{total}$, accounted for $81.3 \pm 15.7\%$ DW of $OC_{total}$. Hence, the contribution of the live seagrass body itself ($OC_{bio}$) was minor ($18.7 \pm 15.7\%$ DW). $OC_{fsed}$ was the major

component of $OC_{sed}$, accounting for $58.3 \pm 14.8\%$ DW of $OC_{total}$; $OC_{csed}$ and $OC_{dead}$ were minor components, accounting for $19.6 \pm 13.7\%$ DW and $3.4 \pm 4.0\%$ DW of $OC_{total}$, respectively.

The average aboveground and belowground biomass in $OC_{bio}$ did not differ significantly between the sites (Fig. 5a) (aboveground biomass, $t = 0.30$, d.f. = 19, $P > 0.05$; belowground biomass, $t = -1.75$, d.f. = 4.67, $P > 0.05$). Belowground biomass accounted for $89.1 \pm 4.4\%$ DW of $OC_{bio}$ (Fig. 5b). The averaged biomasses of aboveground (i.e., leaf) and

belowground (i.e. sheath and rhizome, and root) detritus in $OC_{dead}$ did not differ significantly between the sites (aboveground detritus, $t = 0.60$, d.f. = 7.82, $P > 0.05$; belowground detritus, $W = 28$, $P > 0.05$) (Fig. 5c). The biomass of belowground detritus accounted for $90.8 \pm 12.0\%$ DW of $OC_{dead}$ (Fig. 5d). The biomasses of sheath and rhizome, and root accounted for $65.5 \pm 19.2\%$ DW and $25.3 \pm 16.0\%$ DW of $OC_{dead}$, respectively.

### 3.4 δ¹³C of OC

The average $\delta^{13}C_{sed}$ at the back-reef site ($-12.6 \pm 0.7‰$) was significantly higher than that of the estuarine site ($-16.6 \pm 3.1‰$) ($t = 3.61$, d.f. = 7, $P = 0.008$), and it was also significantly higher than the $\delta^{13}C$ values of algae and coral ($-15.2 \pm 1.9‰$) ($W = 2753$, $P < 0.001$), suspended POM ($-21.9 \pm 1.6‰$) ($t = 15.45$, d.f. = 8, $P < 0.001$), and terrestrial POM ($-28.7 \pm 1.5‰$) ($t = 29.25$, d.f. = 8, $P < 0.001$). However, average $\delta^{13}C_{sed}$ at the back-reef site was significantly lower than $\delta^{13}C$ of seagrass ($-9.2$

$\pm 1.3‰$) ($t = -12.64$, d.f. = 57, $P < 0.001$) (Fig. 6). Average $\delta^{13}C_{sed}$ at the estuarine site did not differ significantly from $\delta^{13}C$ of algae and coral ($W = 457$, $P > 0.05$), but it was significantly higher than $\delta^{13}C$ of both suspended POM ($t = 4.36$, d.f. = 14, $P < 0.001$) and terrestrial POM ($t = 10.05$, d.f. = 14, $P < 0.001$), and significantly lower than $\delta^{13}C$ of seagrass ($t = -6.66$, d.f. = 8, $P < 0.001$). The average $\delta^{13}C$ among fine sediment layers did not differ significantly at either the back-reef site (Wilcoxon signed rank test, Bonferroni adjusted $P > 0.05$) or the estuarine site (paired $t$- test, Bonferroni adjusted $P > 0.05$).



### 3.5 Relationships among biomass, OC mass, and $\delta^{13}C$

At the back-reef site, we found significant correlations between $OC_{sed}$ and DW-based (not carbon-based) biomass ($F_{1,18}$ = 11.63, $P$ = 0.003, $r^2$ = 0.39) (Fig. 7a), $OC_{sed}$ and aboveground biomass ($F_{1,18}$ = 16.38, $P$ < 0.001, $r^2$ = 0.48) (Fig. 7b), $OC_{sed}$ and belowground biomass ($F_{1,18}$ = 10.95, $P$ = 0.004, $r^2$ = 0.38) (Fig. 7c), $OC_{sed}$ and $OC_{dead}$ ($F_{1,18}$ = 4.55, $P$ = 0.047, $r^2$ = 0.20)

(Fig. 7d), and $OC_{sed}$ and $\delta^{13}C_{sed}$ ($F_{1,18}$ = 11.51, $P$ = 0.003, $r^2$ = 0.39) (Fig. 7e). We also found significant correlations between $\delta^{13}C_{sed}$ and belowground biomass ($F_{1,18}$ = 4.68, $P$ = 0.044, $r^2$ = 0.21) (Fig. 7f), and between $\delta^{13}C_{sed}$ and $OC_{dead}$ ($F_{1,18}$ = 13.18, $P$ = 0.002, $r^2$ = 0.42) (Fig. 7g). At the estuarine site, we found significant correlations between $OC_{sed}$ and aboveground biomass ($F_{1,6}$ = 8.18, $P$ = 0.029, $r^2$ = 0.58) (Fig. 7b) and between $OC_{sed}$ and $OC_{dead}$ ($F_{1,6}$ = 6.94, $P$ = 0.039, $r^2$ = 0.54) (Fig. 7d) but not between $OC_{sed}$ and biomass ($F_{1,6}$ = 3.08, $P$ > 0.05, $r^2$ = 0.34) (Fig. 7a), $OC_{sed}$ and belowground biomass (Fig. 7c) ($F_{1,6}$ = 2.94,

$P$ > 0.05, $r^2$ = 0.33), or $OC_{sed}$ and $\delta^{13}C_{sed}$ ($F_{1,6}$ = 0.040, $P$ > 0.05, $r^2$ < 0.01) (Fig. 7e). The slope of the regression line of $OC_{sed}$ against aboveground biomass did not differ significantly between the sites (ANCOVA, $F$ = 1.09, d.f. = 1, $P$ > 0.05) (Fig. 7b), and that of $OC_{sed}$ against $OC_{dead}$ also did not differ significantly between the sites ($F$ = 0.36, d.f. = 1, $P$ > 0.05) (Fig. 7c).

## 4 Discussion

### 4.1 Components of OC stock in seagrass meadows

Our results showed that the sedimentary OC mass ($OC_{sed}$) was the main component of the total organic carbon mass ($OC_{total}$; i.e., all stock components: live and dead above- and below-ground biomass and sediments) at our study sites. If we assume that the density of sedimentary OC is constant to 1-m depth, then we can estimate the relative contribution of $OC_{bio}$ to $OC_{total}$ to be 5.6 ± 3.9% (excluding unvegetated sampling points). This contribution of $OC_{bio}$ to $OC_{total}$, which is the highest among

20 globally compiled data (range, 0.6 ± 0.1% to 2.5 ± 1.4%; Fourqurean et al., 2012), is attributable to the relatively high $OC_{bio}$ and low $OC_{sed}$ at our sites (Fourqurean et al., 2012). The high $OC_{bio}$ was due to the well-developed belowground biomass, which accounted for 90.8 ± 3.9% of $OC_{bio}$ at our sites. This value is also among the highest among globally compiled data (Duarte and Chiscano, 1999). Possible reasons for the exceptional development of belowground biomass include (1) morphological plasticity for resistance to high wave energy (Fonseca and Bell, 1998), which is supported by the low mud

content at our sites compared to that reported by previous studies (Koch, 2001; Serrano et al., 2016a), and (2) nutrient limitation, which can lead to more allocation of biomass to belowground parts to enable the plant to acquire nutrients in deeper sediment layers (Lee et al., 2007). The low $OC_{sed}$ may be attributable to (1) high wave energy in association with increased OC lability due to the low specific surface area of sediments (Miyajima et al., 2017) and (2) the low gross primary production/respiration (P/R) ratio in this geographical region (Duarte et al., 2010).

Belowground detritus (i.e., sheath and rhizome, and root) was the major component of $OC_{dead}$, accounting for 90.8 ± 12.0% of $OC_{dead}$ at our sites. This result is consistent with a previous report on *Cymodocea nodosa* (Cebrian et al., 2000) and



suggests that belowground detritus is more easily stored in the sediment than aboveground detritus. A mechanism supporting this hypothesis might be either (1) a higher belowground biomass and an associated higher supply of seagrass detritus or (2) higher recalcitrance of belowground detritus. Here, a higher supply is more likely because at our sites the belowground biomass is among the highest reported values for each species (Duarte and Chiscano, 1999), although the reported aboveground/belowground production ratio of *T. hemprichii* and *E. acoroides* varies among studies (Duarte et al., 1998; Duarte and Chiscano, 1999; Erftemeijer, Osinga, and Mars, 1993). Higher recalcitrance is also possible; Holmer and Olsen (2002) reported that during a 43 day decomposition experiment, *E. acoroides* rhizomes did not lose weight, whereas buried leaves lost $80.3 \pm 4.2\%$ of their weight. Also, Fourqurean and Schrlau (2003) showed that only $5 \pm 2\%$ of *Thalassia testudinum* leaves, but $49 \pm 6\%$ of *T. testudinum* rhizomes, remained after 348 days of decomposition.

## 4.2 Mechanism of the OC supply to sediment

$OC_{sed}$ was significantly and positively correlated with aboveground biomass at both sites (Fig. 7b) and to belowground biomass at the back-reef site (Fig. 7c). This result is contrary to the finding of most previous studies that there is no relationship between biomass and OC (Kennedy et al., 2004, 2010; Howard et al., 2017; but cf. Samper-Villarreal et al., 2016). This contrary result may be due to our data collection strategy of (1) sympatric sampling of all stock components (live and dead above- and belowground biomass and sediments) in intact cores, and (2) selection of sampling points aiming at controlling for variables other than seagrass biomass (i.e., mud content, wave height, and the amount of allochthonous OC inputs were relatively homogenous among points). Several mechanisms can plausibly explain the positive relationship between seagrass biomass and $OC_{sed}$, including (1) trapping of suspended OC (both allochthonous and autochthonous OC) by seagrass leaves, and (2) the direct supply of belowground seagrass-derived autochthonous OC. If it is assumed that the suspended OC settling on the sediment surface is spatially homogeneous in nature (quality) and that the contribution of trapped OC is larger than that of directly supplied OC, then $\delta^{13}C_{sed}$ should be constant regardless of the aboveground biomass and its associated trapping capacity. However, $OC_{sed}$ was significantly and positively correlated with $\delta^{13}C_{sed}$ (Fig. 7e), and the average $\delta^{13}C$ of $OC_{sed}$ was significantly higher than $\delta^{13}C$ values of allochthonous OC (algae and corals, suspended POM, and terrestrial POM) at the back-reef site (Fig. 6). Furthermore, $OC_{sed}$ was positively correlated with $OC_{dead}$ at both sites (Fig. 7d), and the main component of $OC_{dead}$ was belowground detritus (Fig. 5d). Taken together, these results suggest that directly supplied seagrass-derived OC was mainly from the belowground detritus. The positive correlations between $\delta^{13}C_{sed}$ and belowground biomass (Fig. 7f) and between $\delta^{13}C_{sed}$ and $OC_{dead}$ (Fig. 7g) at the back-reef site also support this mechanism. From these lines of evidence, we conclude that the direct supply of recalcitrant belowground seagrass detritus is a major mechanism of $OC_{sed}$ enrichment at the back-reef site (Fig. 8). Although we inferred that a direct autochthonous OC supply from belowground biomass is the major mechanism of $OC_{sed}$ enrichment, suspended allochthonous OC may also have been supplied from the water column at the

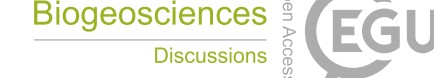

back-reef site, as has been reported elsewhere (Kennedy et al., 2010), because the average $\delta^{13}C_{sed}$ at the back-reef site was significantly lower than $\delta^{13}C$ of seagrass (Fig. 6).

The seagrass-derived OC increase according to the development of the seagrass meadows at the back-reef site (Fig. 7a and Fig. 7e) suggests that seagrass meadows are autotrophic and the time since seagrass colonization is longer. This inference is consistent with a previous report that net primary production (NPP) at the back-reef site is higher where the seagrass cover is high (cover 91.7%; NPP 68.14 mmol C m$^{-2}$ d$^{-1}$) than where the seagrass cover is low (cover 55.1%; NPP 34.20 mmol C m$^{-2}$ d$^{-1}$) (Nakamura and Nakamori, 2009). It is also possible that seagrass mortality increases with time since colonization, leading to an increase in dead plant structures (Cebrian et al., 2000).

At the estuarine site, $OC_{sed}$ increased with increasing aboveground seagrass biomass (Fig. 7b), but it did not increase with increasing belowground seagrass biomass (Fig. 7c), indicating that trapping of suspended OC by seagrass leaves surpassed the direct supply of belowground seagrass-derived OC (Fig. 8). However, $OC_{dead}$ was significantly and positively correlated with $OC_{sed}$ (Fig. 7d), indicating that direct supply also contributed to $OC_{sed}$ enrichment at the site. A plausible mechanism explaining the hypothesized dominance of suspended OC trapping is a lower belowground turnover rate (i.e., the production/biomass ratio) at the estuarine site than at the back-reef site. Because $OC_{sed}$ was not significantly different between the sites and directly supplied seagrass-derived OC was the major component of $OC_{sed}$ at the back-reef site and only a minor component at the estuarine site, the capacity of the estuarine site to directly supply belowground seagrass-derived OC to the sediment was lower than that of the back-reef site (Fig. 8). Moreover, given that the directly supplied amount is determined by two factors, the belowground biomass and its turnover rate, and that the belowground biomass was not significantly different between the sites (Fig. 5a), we anticipate that a difference in the belowground turnover rate was responsible for the difference in the direct supply contribution between the sites. Another possible explanation for the inferred difference is that the absolute input of allochthonous OC was higher at the estuarine site than at the back-reef site. The slope of the regression between aboveground biomass and $OC_{sed}$ was not significantly different between the sites (Fig. 7b), which suggests that the trapping ability for autochthonous and allochthonous OC was not different between the sites. However, the fact that $OC_{sed}$ was not significantly different between the sites (Fig. 4) together with the apparently minor direct belowground supply at the estuarine site implies that the contribution of OC from the water column to $OC_{sed}$ was larger at the estuarine site. Moreover, the fact that average $\delta^{13}C_{sed}$ was significantly lower at the estuarine site than at the back-reef site (Fig. 6) would support a major role of allochthonous OC from the water column in $OC_{sed}$ enrichment. The effect of particle trapping by seagrasses is reported to be enhanced particularly in particle-poor waters (Duarte et al., 1999). Thus, trapping is likely to be an important mechanism especially at sites with particle-poor water such as coral reef sites.

**Biogeosciences** Open Access
Discussions
EGU

## 5 Conclusion

Using our data collection strategy, namely, sympatric sampling in intact cores of live and dead seagrass bodies and sediments and analyses of the organic carbon mass and stable carbon isotope composition of all components of the cores, we successfully demonstrated the pathways of sedimentary OC enrichment in seagrass meadows and showed that the contributions of both a
5   direct supply of seagrass-derived OC by belowground production and particle trapping are important, although the latter is generally assumed as the main mechanism of OC enrichment in seagrass meadows compared with the bare sediment sites. Our results indicate that it is critical to consider both below- and aboveground biomass productivity in addition to the morphological complexity of seagrass meadows as factors controlling OC, and that identifying the mechanism of enrichment is important for improving OC stock estimation and reducing the uncertainty in global blue-carbon estimates.

**Authors contribution**

Toko Tanaya, Hajime Kayanne, and Tomohiro Kuwae conceived the idea; Toko Tanaya, Kenta Watanabe, and Tomohiro Kuwae designed the methodology; Toko Tanaya, Kenta Watanabe, Shoji Yamamoto, and Chuki Hongo collected the samples and data; Toko Tanaya and Kenta Watanabe performed sample analyses; Toko Tanaya and Tomohiro Kuwae led the writing
of the manuscript. All authors contributed critically to the drafts and gave final approval for publication.

**Competing interests**

The authors declare that they have no conflict of interest.

**Acknowledgements**

We thank R. Tada for helpful comments; A. Watanabe, H. Takamiyagi, and E. Shimabukuro for the field sampling; K. Sakihara and A. Okuno for chemical analyses; S. Ogihara, T. Muranaka, and N. Koh for sample preparation; and S. Kaku for developing the box corer. This study was partly supported by a Grant-in-Aid for Challenging Exploratory Research (No. 26630251) from the Japan Society for the Promotion of Science (JSPS), and by the Strategic R&D Area Project (S-14) of the Environmental
Research and Technology Development Fund of the Ministry of the Environment, Japan.



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





**Table 1: Organic carbon contents and dry weights of each component of living biomass at the back-reef and estuarine sites.**

| | Back reef | | Estuary | |
|---|---|---|---|---|
| | %OC (%DW) | Dry weight (g m$^{-2}$) | %OC (%DW) | Dry weight (g m$^{-2}$) |
| | mean ± SD ($n$) | mean ± SD ($n$) | mean ± SD ($n$) | mean ± SD ($n$) |
| Aboveground biomass | 38.47 ± 3.06 (39) | 74 ± 45 (16) | 35.95 ± 2.28 (20) | 70 ± 34 (5) |
| Belowground biomass | 31.35 ± 2.93 (20) | 675 ± 450 (16) | 30.38 ± 2.55 (13) | 1354 ± 847 (5) |



**Table 2: Organic carbon content, δ¹³C, and dry density of each of sediment and dead plant component at the back-reef and estuarine sites.**

| | Back reef | | | Estuary | | |
|---|---|---|---|---|---|---|
| | Organic carbon | | Dry density | Organic carbon | | Dry density |
| | %OC (% DW) mean ± SD (*n*) | δ¹³C (‰ vs. VPDB) mean ± SD (*n*) | (g cm⁻³) mean ± SD (*n*) | %OC (% DW) mean ± SD (*n*) | δ¹³C (‰ vs. VPDB) mean ± SD (*n*) | (g cm⁻³) mean ± SD (*n*) |
| Fine sediment | 0.37 ± 0.13 (60) | −12.8 ± 0.8 (60) | 0.89 ± 0.30 (60) | 0.42 ± 0.20 (24) | −17.4 ± 3.6 (24) | 0.76 ± 0.29 (24) |
| Coarse sediment | 0.32 ± 0.13 (20) | −12.8 ± 1.1 (20) | 0.29 ± 0.15 (20) | 0.26 ± 0.08 (8) | −15.9 ± 1.5 (8) | 0.48 ± 0.14 (8) |
| Dead leaf | 24.80 ± 3.07 (3) | −8.9 ± 0.6 [a] (5) | 0.00 ± 0.00 (20) | 23.31 ± 3.86 [b] (3) | −9.3 ± 0.2 [a] (5) | 0.00 ± 0.00 (8) |
| Dead sheath and rhizome | 21.29 ± 4.07 (3) | −8.9 ± 0.6 [a] (5) | 0.00 ± 0.00 (20) | 27.52 ± 1.75 [b] (3) | −9.3 ± 0.2 [a] (5) | 0.00 ± 0.00 (8) |
| Dead root | 19.25 ± 1.67 (3) | −8.9 ± 0.6 [a] (5) | 0.00 ± 0.00 (20) | 19.94 ± 5.89 [b] (3) | −9.3 ± 0.2 [a] (5) | 0.00 ± 0.00 (8) |

[a]**Total of sheath and rhizomes, and root.**

[b]**At one sampling point (FS1) where the dominant species was different, the values were dead leaf, 25.77%; dead sheath and rhizome, 19.05%; and dead root, 19.21%.**

**Figure Captions**

Figure 1: Study sites. (a) Study site location on Ishigaki Island, Japan. Sampling points at (b) the back-reef

and (c) the estuarine site. At the back-reef site, the circle indicating the southernmost vegetated sampling

point actually represents a cluster of six sampling points.

Figure 2: The newly developed box corer and a sampled core. Schematic diagrams of (a) a cross section of

a core and (b) the design of the corer. (c) Photograph of a core from the back-reef site. The dominant

seagrass species is *Thalassia hemprichi*.

10    Figure 3: Calculation of total OC mass ($OC_{total}$; g C m$^{-2}$) in the top 0.15-m layer.

Figure 4: OC mass ($OC_{bio}$, $OC_{dead}$, $OC_{csed}$, $OC_{fsed}$, $OC_{sed}$, and $OC_{total}$) at (a) the back-reef site and (b) the

estuarine site. Boxes show the 25% and 75% quantiles; horizontal bands inside the box are median values;

whiskers show maximum and minimum values; and the open circle is an outlier.

Figure 5: (a) $OC_{bio}$ (sum of aboveground and belowground biomass) (g C m$^{-2}$); (b) contribution of

belowground biomass to $OC_{bio}$ (%); (c) $OC_{dead}$ (sum of above- and belowground detritus (g C m$^{-2}$); and (d)



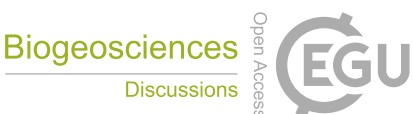

contribution of belowground detritus to $OC_{dead}$ (%). Boxes show the 25% and 75% quantiles; horizontal bands inside the box are median values; whiskers show maximum and minimum values; and open circles show outliers.

5    Figure 6: $\delta^{13}C_{sed}$ at each site and the $\delta^{13}C$ values of potential sources of OC of $\delta^{13}C_{sed}$ (means ± SE).

Figure 7: Relationships at the back-reef (blue) and estuarine (orange) sites between $OC_{sed}$ and (a) biomass (g C m$^{-2}$), (b) aboveground biomass (g C m$^{-2}$), (c) belowground biomass (g C m$^{-2}$), and (d) $OC_{dead}$ (g C m$^{-2}$), and between (e) $OC_{sed}$ and $\delta^{13}C_{sed}$, (f) $\delta^{13}C_{sed}$ and belowground biomass, and (g) $\delta^{13}C_{sed}$ and $OC_{dead}$.

Figure 8: Proposed mechanisms of OC stock enrichment at our study sites. At the back-reef site dominated by *Thalassia hemprichii*, direct supply of recalcitrant belowground seagrass detritus is a major pathway of $OC_{sed}$ enrichment. At the estuarine site dominated by *Enhalus acoroides*, trapping of suspended autochthonous and allochthonous OC is the major pathway of OC enrichment. A difference in the turnover

15    rate of belowground biomass likely caused the major mechanism of OC stock enrichment to differ between the sites.



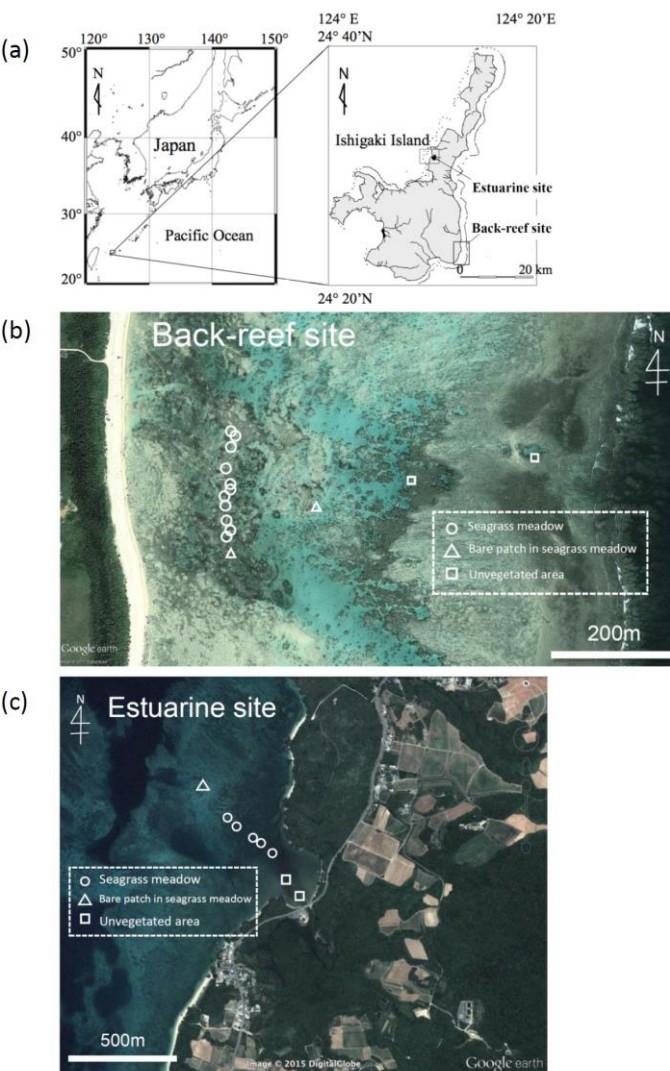

Figure 1: Study sites. (a) Study site location on Ishigaki Island, Japan. Sampling points at (b) the back-reef and

(c) the estuarine site. At the back-reef site, the circle indicating the southernmost vegetated sampling point

actually represents a cluster of six sampling points.



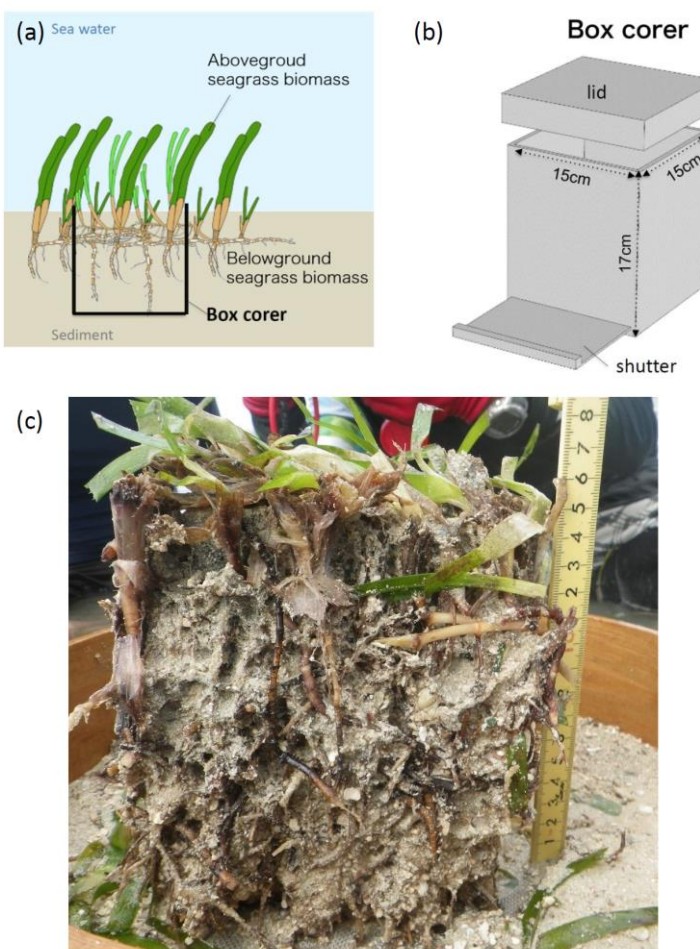

**Figure 2: The newly developed box corer and a sampled core. Schematic diagrams of (a) a cross section of a core**

**and (b) the design of the corer. (c) Photograph of a core from the back-reef site. The dominant seagrass species**

**is *Thalassia hemprichi*.**





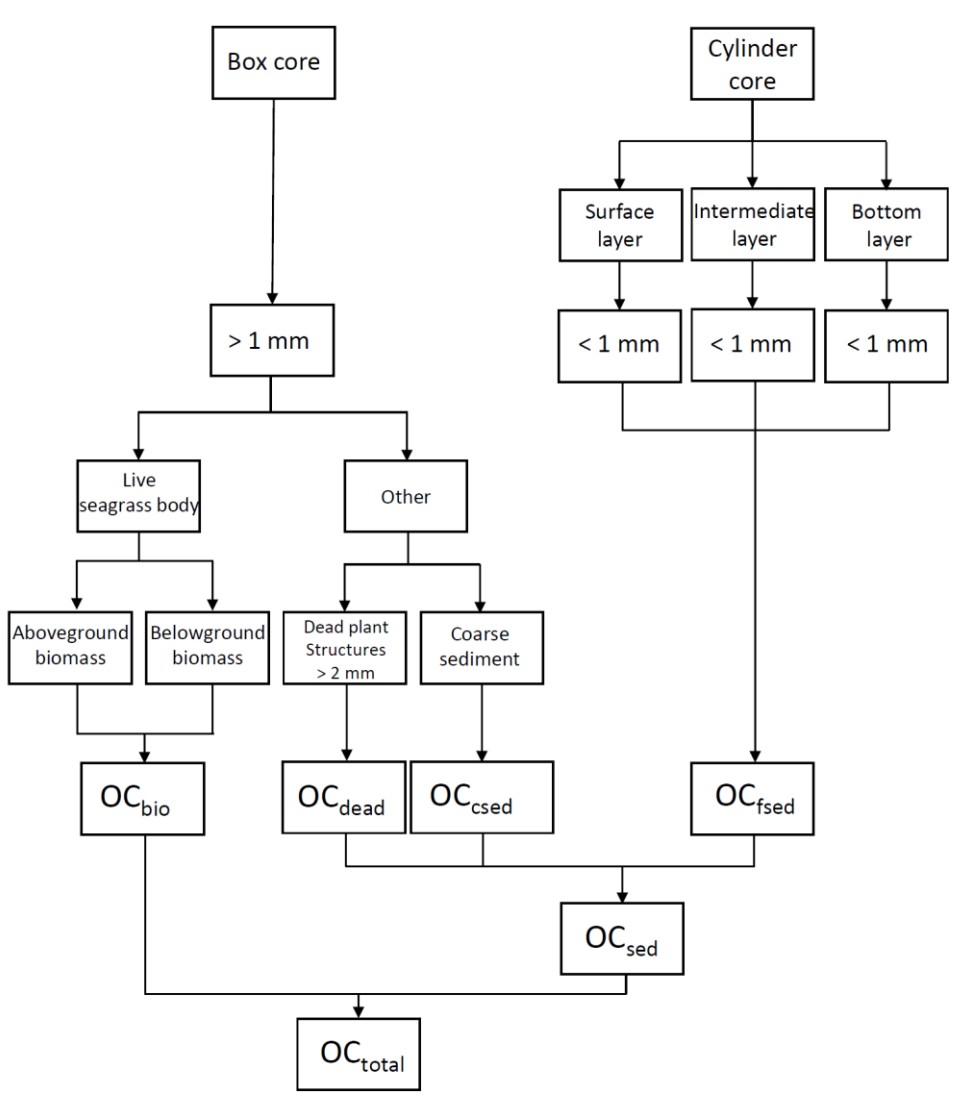

**Figure 3: Calculation of total OC mass (OC$_{total}$; g C m$^{-2}$) in the top 0.15-m layer.**



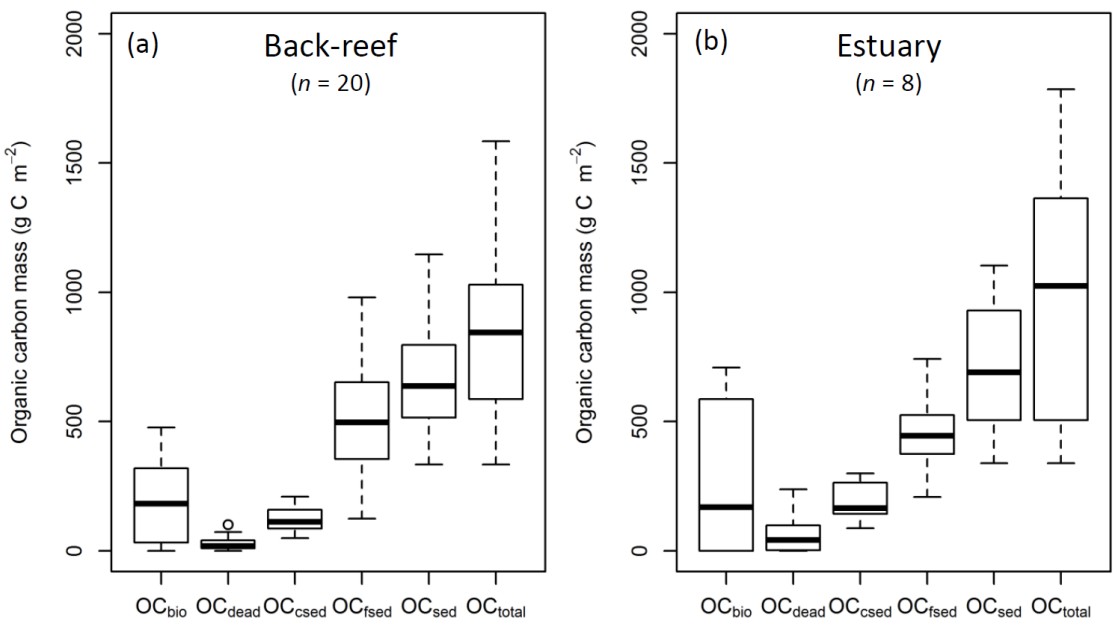

**Figure 4: OC mass ($OC_{bio}$, $OC_{dead}$, $OC_{csed}$, $OC_{fsed}$, $OC_{sed}$, and $OC_{total}$) at (a) the back-reef site and (b) the estuarine site. Boxes show the 25% and 75% quantiles; horizontal bands inside the box are median values; whiskers show maximum and minimum values; and the open circle is an outlier.**



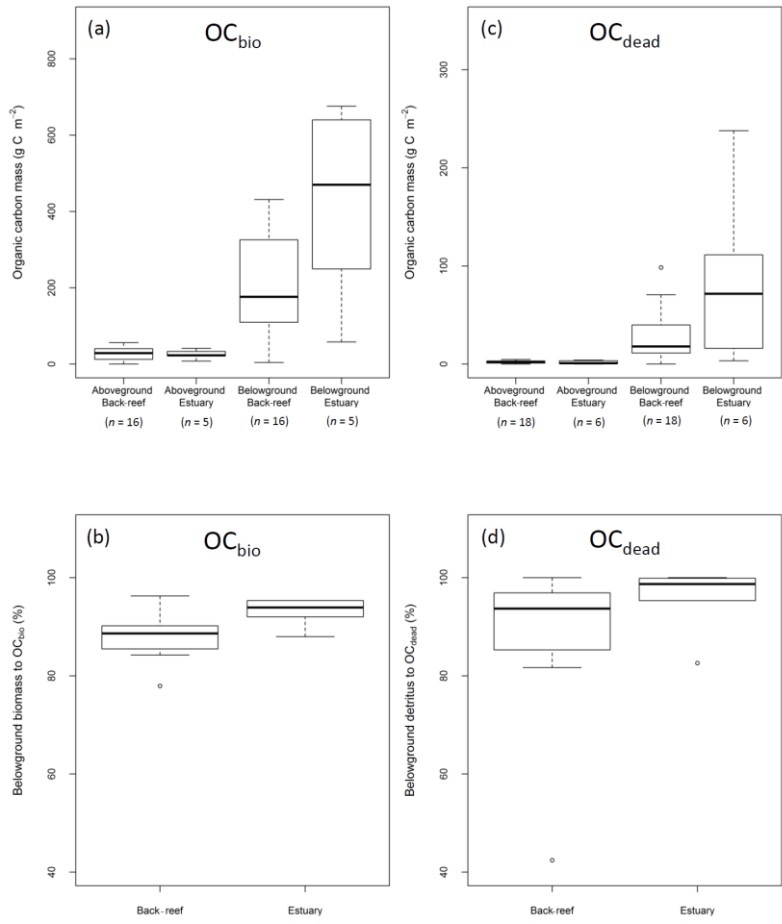

**Figure 5:** (a) OC$_{bio}$ (sum of aboveground and belowground biomass) (g C m$^{-2}$); (b) contribution of belowground

biomass to OC$_{bio}$ (%); (c) OC$_{dead}$ (sum of above- and belowground detritus (g C m$^{-2}$); and (d) contribution of

belowground detritus to OC$_{dead}$ (%). Boxes show the 25% and 75% quantiles; horizontal bands inside the box

5    are median values; whiskers show maximum and minimum values; and open circles show outliers.



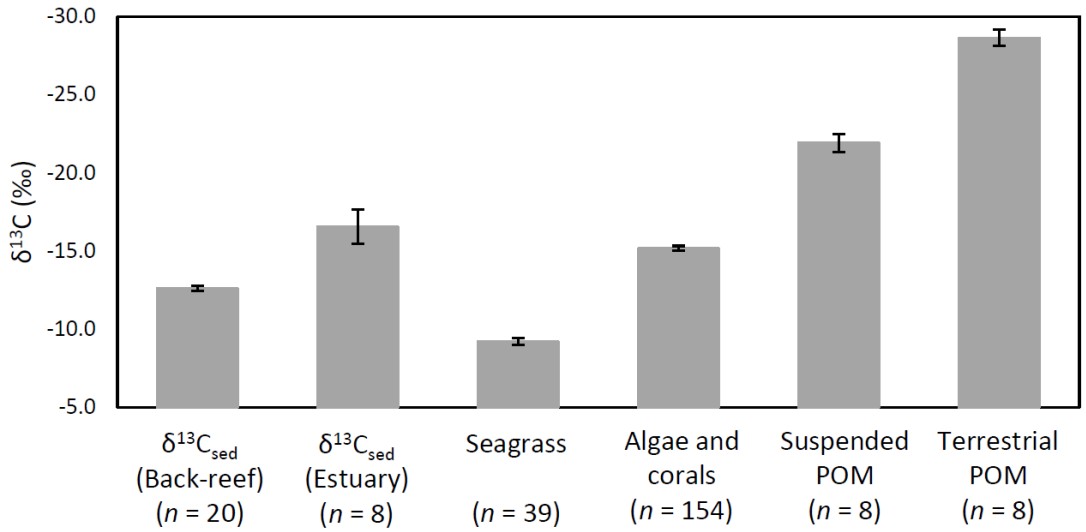

**Figure 6: $\delta^{13}C_{sed}$ at each site and the $\delta^{13}C$ values of potential sources of OC of $\delta^{13}C_{sed}$ (means ± SE).**



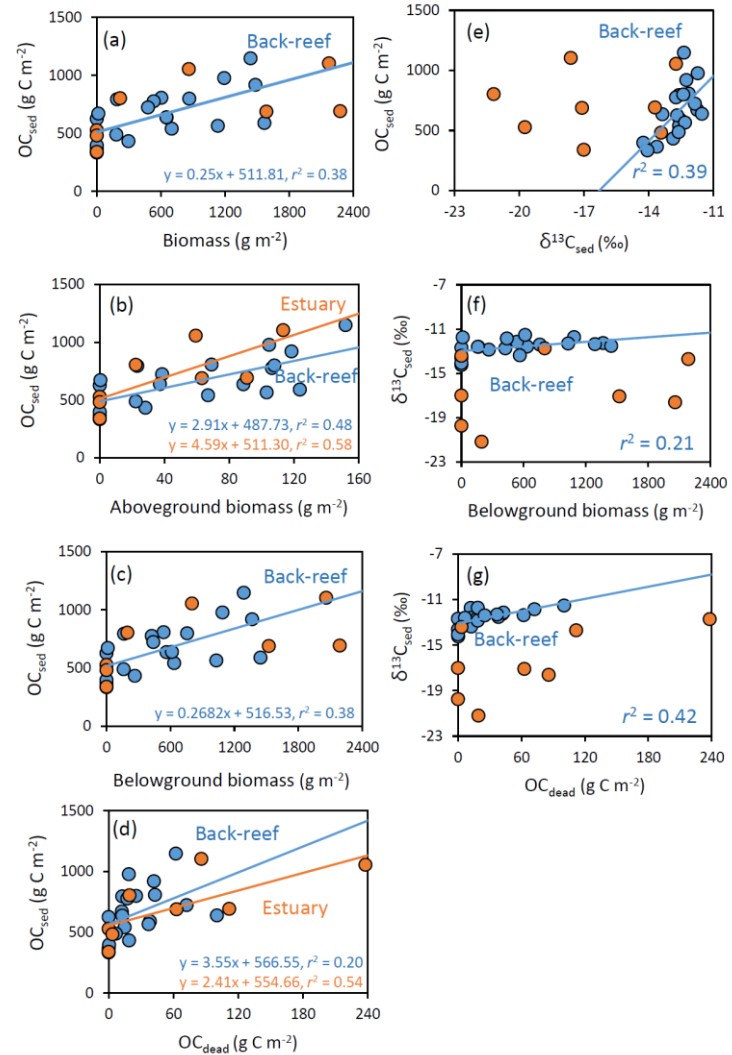

**Figure 7: Relationships at the back-reef (blue) and estuarine (orange) sites between OC$_{sed}$ and (a) biomass (g C m$^{-2}$), (b) aboveground biomass (g C m$^{-2}$), (c) belowground biomass (g C m$^{-2}$), and (d) OC$_{dead}$ (g C m$^{-2}$), and between (e) OC$_{sed}$ and δ$^{13}$C$_{sed}$, (f) δ$^{13}$C$_{sed}$ and belowground biomass, and (g) δ$^{13}$C$_{sed}$ and OC$_{dead}$.**

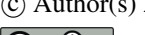



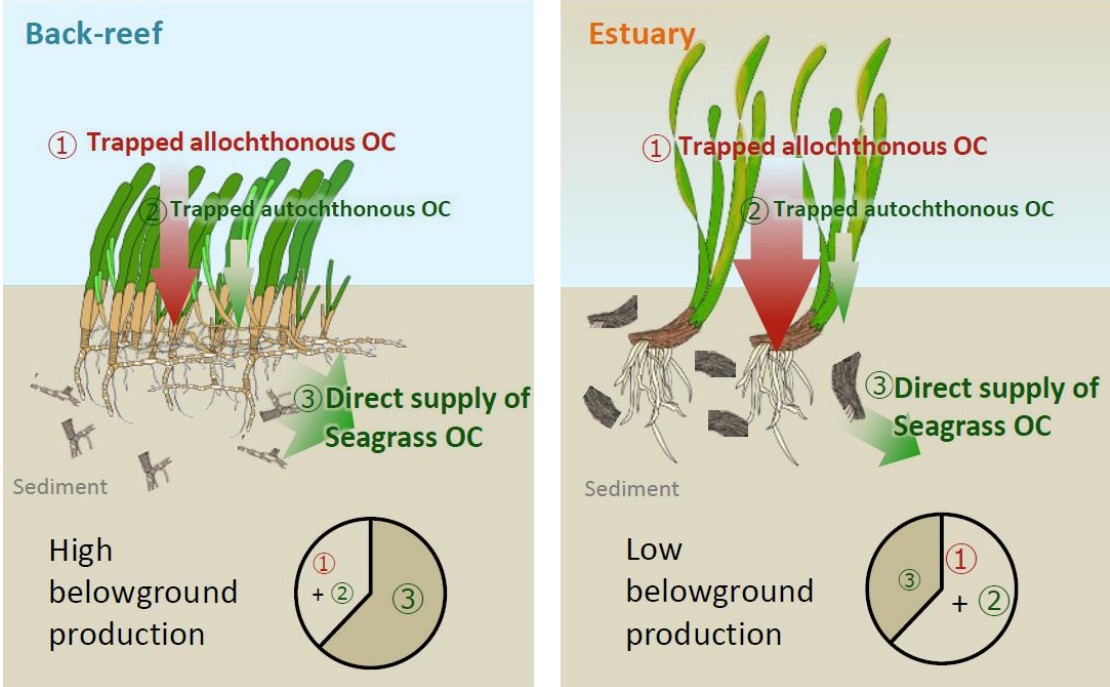

**Figure 8: Proposed mechanisms of OC stock enrichment at our study sites. At the back-reef site dominated by**

***Thalassia hemprichii*, direct supply of recalcitrant belowground seagrass detritus is a major pathway of $OC_{sed}$**

**enrichment. At the estuarine site dominated by *Enhalus acoroides*, trapping of suspended autochthonous and**

5      **allochthonous OC is the major pathway of OC enrichment. A difference in the turnover rate of belowground**

**biomass likely caused the major mechanism of OC stock enrichment to differ between the sites.**