# Peer review of "Contributions of the direct supply of belowground seagrass detritus and trapping of suspended organic matter to the sedimentary organic carbon stock in seagrass meadows"

_Biogeosciences, 2017_

## Referee Comment (RC1) · Anonymous Referee #1 · 5 Jan 2018

The paper was well executed and written and presented novel data on seagrass carbon dynamics. Particularly, this paper fills in a much needed gap on tropical blue carbon ecosystems and the contribution of belowground biomass (esp. sheathes) to carbon stocks, the latter often erroneously overlook or lumped in as the sediment carbon stock. It would be interesting to expand on this study by looking at similar variables at deeper depths so that (a) it is comparable to global studies that look at 30-100 cm depths, and (b) we can understand better the long-term contributions of seagrass and allochthonous OC were living biomass isn't present and detritus has been processed

more by microbial remineralization. There are some concerns about the lumping of different vegetation types into a site average, but otherwise these are minor revisions.

Abstract • Lines 3, 11, 13: What is meant by enrichment? Looking at the next sentences, 'accumulation' may be a more accurate term. Change throughout the manuscript. • Line 5: 'bodies' is an uncommon term for seagrasses and should be 'plants' or 'biomass' here and throughout the manuscript • It will be helpful to describe what species of seagrass are being studied in the abstract. • Line 16: no need to hyphenate blue carbon. Change throughout the manuscript as well.

Introduction • Lines 17-30: Consider Trevathan-Tackett et al. 2017 as a specific review of seagrass recalcitrance and the potential for contributing to OC stocks (doi: 10.3389/fpls.2017.00925); it will also be useful in the first section of the discussion. Also consider new research on providence of OC in seagrass meadows using eDNA: Reef et al 2017 doi: 10.1002/lno.10499

Methods • How are you considering leaf detritus in these sediment measurements/calculations? In sections 2.2 it says it's a part of the dead plant structures but not in the calculations. Is it assumed that 100% of the surface leaf detritus is exported and not buried? • Why is the Cfsed calculation multiplied by 1/3 (eqs. 6 & 8)? • How do equations 7 and 8 relation to traditional mixing model methods to look at OC providence? Were the end-members (seagrass, POM, algae/coral, terrestrial) taken into account? It seems a waste not to use this stable isotope to quantitatively obtain OC contribution values.

Results • Since section 3.2 only has one sentence, I'd suggest adding it to the next OC section • One suggestion is to make a supplementary table(s) for the statistics. This would make reading the text easier. • Where are the data on the differences between vegetated, unvegetated and bare OC stocks and fractions? This will be very important in the interpretation of OCbio and OCdead. This will give better resolution into the differences within and between back-reef and estuary regions • What about

correlations between living AG:BG?

Discussion • Page 10, Lines 17-19: NO, we cannot assume constant to 1-m depth. There are important processes that affect OC down core, most notably the reduction on living biomass with depth, change in bulk density and microbial remineralization, so there is absolutely no meaning to the OCbio to OCtotal estimate. Please remove this sentence and calculation and find another more robust way to compare the OCbio data to previous literature • Second paragraph: Anoxic sediments that generally reduce decomposition rates also can lead to higher preservation of OC

Figures Figure 1 is low quality and fuzzy and thus hard to read Figure 3: please define the abbreviations in the caption.

---

## Short Comment (SC1) · 14 Feb 2018

Rozaimi, M. and Hamdan N. H.

School of Environmental and Natural Resource Sciences, Universiti Kebangsaan Malaysia, UKM Bangi, Selangor 43600, Malaysia

—

The study by Tanaya et al. reports findings in the context of blue-carbon science,

specifically as a representation for the Indo-Pacific region. The authors demonstrated meticulous planning for the study and the manuscript is generally well-written. Our group is similarly involved in blue carbon studies and we draw some corollary between this study and our findings. In addition, we suggest some recommendations that may improve the authors' present and future outlook in this field.

One of the highlights of this study is the argument on the contribution of biomass-derived organic carbon (OC bio) to the organic carbon pool (OC total) as the highest globally (P2L8). The data is presented in percentages (i.e. 19% OC bio and 81% OC sed) rather than the actual organic carbon stocks. It may be apt to complement such comparisons with actual global stock values (in equivalent measures as grams C per meter squared or megagrams C per hectare). They then rounded off their study by stating below-ground biomass is a driver for sediment OC storage (P2L14-15). It may hold true for this specific study, which is represented by findings from two sites. The authors rightly indicated past studies (e.g. Kennedy et al., 2004; 2010 and Howard et al. 2017 – P11L14) showed no relationships between seagrass biomass and sediment OC stocks. This is consistent with our recent study as well (Rozaimi et al. 2017). However, our other studies suggest otherwise whereupon biomass is indeed important in driving sediment OC stocks (Serrano et al. 2016; Rozaimi et al. 2013). Tanaya et al. provided possible explanations on why they had different outcomes (P11) but alternatively, it may be plausible that their study sites may simply have exceptional sediment OC storage characteristics compared to other Indo-pacific seagrass meadows.

Further to the above, it has to be clearly noted this study reports findings from surficial sediments (up to 16 cm depth: P5L5). This depth is within the range of vertical rhizomal growths for Indo-pacific seagrass rhizomes (especially T. hemprichii). So clearly autochthonous inputs play an important role in retaining seagrass-derived OC within this depth layer. However, the context of the authors' findings within 15 cm sediment depths up-scaled to 1 m, on the assumption that sediment OC density is constant (P10L18) may be too broad an assumption. In our published results (Rozaimi et al.

2017), we found variability in surficial downcore OC content (up to 30 cm sediment depth, albeit as %OC) as well as changing $\delta$13C sediment signatures with increasing sediment depth. In other studies (Rozaimi et al. in preparation), we did not find consistency in downcore OC content or OC density in cores up to 1 m. Conventionally, the scaling-up approach is employed (and admittedly we have used scaling-up approaches to model sediment OC stocks up to 1 m) to contextualise findings relative to regional and global estimates as that in Fourqurean et al. (2012). The authors' assumption in this regard may be corroborated if other evidence can be presented to support the notion of past seagrass occurrences in their study site (re: Serrano et al. 2016; Belshe et al. 2017). Or simply, such investigations may be room for improvements in the authors' future work.

On a final note, it is particularly interesting the authors have data (though not apparently analysed as yet) that can be used in stable isotope mixing models. Mixing models have been increasingly used to account for the contributions of seagrass derived-OC to bulk sediment organic pool and could thus offer alternative insights to the authors' findings. We do wonder how the authors' approach in this study hold up compared to approaches such as stable isotope analysis in R (SIAR; e.g Watanabe and Kuwae 2015; Rozaimi et al. 2017) or eDNA approaches (Reef et al 2017). The lack of reference to SIAR, at least, is somewhat peculiar since there are co-authors in this current study, who are familiar with SIAR (i.e. Watanabe and Kuwae 2015).

Overall, we view this study as interesting and may well be citable in future blue carbon endeavours.

—

General technical comments:

Seagrass "bodies" is a peculiar term to use

On the use of "enrichment": conventionally, communications in this regards may construe the presence of higher quantity of 13-C atoms (i.e. enriched samples) relative to non-enriched samples. In the text, readers may find some confusion on whether the authors refer to 13-C enrichment, or simply linguistic reference to higher amounts of a particular entity.

P4L14-22: Content more suited in the Introduction section

P11L14: A word missing after OC (perhaps OC stocks?)

P21 Table 2: On data entries as 0.00 $\pm$ 0.00: do these data refer to nil values, or data values less than 0.001?

P28 Figure 5: Axis labels are too small

—

Acknowledgements

We acknowledge the research university grant UKM-GGPM-2016-033, which supported our work in assessing the capacity of tropical seagrass meadows for carbon sequestration.

—

References

Belshe, E. F., Mateo, M. A., Gillis, L., Zimmer, M. and Teichberg, M.: Muddy Waters: Unintentional consequences of Blue Carbon research obscure our understanding of organic carbon dynamics in seagrass ecosystems, Front. Mar. Sci., 4, 1–9, doi:10.3389/fmars.2017.00125, 2017.

Fourqurean, J. W., Duarte, C. M., Kennedy, H., Marba, N., Holmer, M., Mateo, M. A., Apostolaki, E. T., Kendrick, G. A., Krause-Jensen, D., McGlathery, K. J. and Serrano, O.: Seagrass ecosystems as a globally significant carbon stock, Nat. Geosci., 5, 505–509, doi:10.1038/ngeo1477, 2012.

Howard, J. L., Creed, J. C., Aguiar, M. V. P. and Fouqurean, J. W.: CO2 released by carbonate sediment production in some coastal areas may offset the benefits of seagrass "Blue Carbon" storage, Limnol. Oceanogr., 160–172, doi:10.1002/lno.10621, 2017.

Kennedy, H., Gacia, E., Kennedy, D. P., Papadimitriou, S. and Duarte, C. M.: Organic carbon sources to SE Asian coastal sediments, Estuar. Coast. Shelf Sci., 60, 59–68, doi:DOI: 10.1016/j.ecss.2003.11.019, 2004.

Kennedy, H., Beggins, J., Duarte, C. M., Fourqurean, J. W., Holmer, M., Marba, N. and Middelburg, J. J.: Seagrass sediments as a global carbon sink: isotopic constraints, Global Biogeochem. Cycles, 24, GB4026, doi:10.1029/2010GB003848, 2010.

Reef, R., Atwood, T. B., Samper-Villarreal, J., Adame, M. F., Sampayo, E. M. and Lovelock, C. E.: Using eDNA to determine the source of organic carbon in seagrass meadows, Limnol. Oceanogr., 62, 1254–1265, doi:10.1002/lno.10499, 2017.

Rozaimi, M., Serrano, O. and Lavery, P. S.: Comparison of carbon stores by two morphologically different seagrasses, J. R. Soc. West. Aust., 96, 81–83, 2013.

Rozaimi, M., Fairoz, M., Hakimi, T. M., Hamdan, N. H., Omar, R., Ali, M. M. and Tahirin, S. A.: Carbon stores from a tropical seagrass meadow in the midst of anthropogenic disturbance, Mar. Pollut. Bull., 119, 253–260, doi:https://doi.org/10.1016/j.marpolbul.2017.03.073, 2017.

Serrano, O., Ricart, A. M., Lavery, P. S., Mateo, M. A., Arias-Ortiz, A., Masque, P., Rozaimi, M., Steven, A. and Duarte, C. M.: Key biogeochemical factors affecting soil carbon storage in Posidonia meadows, Biogeosciences, 13, 4581–4594, doi:doi:10.5194/bg-13-4581-2016, 2016.

Watanabe, K. and Kuwae, T.: How organic carbon derived from multiple sources contributes to carbon sequestration processes in a shallow coastal system? Global Change Biol., 21, 2612–2623, doi:10.1111/5 gcb.12924, 2015.

---

## Referee Comment (RC2) · Anonymous Referee #2 · 18 Mar 2018

General comments

This study aims to assess the mechanisms constraining organic carbon storage at two sites in Japan colonised by seagrass meadows quantifying the different pools of organic carbon that contribute to sediment organic carbon stock in seagrass sediments (and unvegetated sediments). The study demonstrates that seagrass structure and detritus constrain sediment organic carbon stores at the study sites. The manuscript is well written. However, I have some comments that I list in detail below.

Specific comments

Introduction. Page 3, line 33/Page 4, line 1. It is not clear in this sentence if the authors mean organic carbon or carbonate of calcareous organisms.

Introduction. I suggest to re-write the last paragraph of the introduction to highlight the novel aspects of the study.

Methods. Study site. The first paragraph could be moved to the introduction.

Methods, page 4 last paragraph and Fig. 1. The location of the river mouth of Todoroki River relative to the sampling site is not clearly shown in the figure. This prevents to understand why the terrestrial input in this site is low. Similarly, the location of the small river discharging into the estuary is not clear in the image.

Methods. Page 5. It is not clear the type of organic material included in the fraction OCcsed. If it contained the carbonate from skeletons of corals, foraminifera, and other calcareous organisms it should not be considered in the organic carbon pool.

Methods. Page 5. Line 24. "We merged dead plant structures attached to live seagrass bodies into OCbio". How much did dead plant structures attached to living biomass weight? How much was it in comparison to mass of the seagrass dead compartment? Could this affect the OC results across compartments?

Methods. Page 5, last paragraph. At each site, samples were collected in vegetated, unvegetated patches within the meadows and bare sediment. However the results in the box plots (Figs. 4 and 5) are presented per site, without indicating if they correspond to vegetated, unvegetated patches or bare sediment. I think it would be relevant to present these results indicating if the sediments were vegetated or not.

Table 2. In this table the density of dead plant material is $0.00 \pm 0.00$ g cm-3. I believe that these components did have some dry density but lower than 0.00 g cm-3. I order to be able to provide their dry density, the units could be expressed in mg cm-3.

Discussion. How much was the OC sediment stock at the studied seagrass meadows and at the bare sites? How do the OC stocks in the seagrass sediments found in this study compare with global seagrass OC sed stocks?

Discussion. What is the contribution of the different potential OC sources (seagrass, algae, corals, suspended POM and terrestrial POM) to OC in the sediment at both sites (and discriminating between vegetated and bare sediment)? The fraction of the different sources to the compartments of coarse and fine sediment could be estimated using mixing models. These estimates could be incorporated in a revised Fig. 8.

Conclusions. Kennedy et al 2010 and several other papers demonstrate that the contribution of particle trapping and seagrass material to sediment organic carbon widely varies across seagrass meadows, from meadows where allochthonous carbon is the main source to others where the sediment organic carbon pool is dominated by seagrass material. Therefore, there is evidence in the literature that seagrass carbon can be an important source to sediment organic carbon.

Minor comments Abstract- line 7. It should say that the stable carbon isotope ratio was measured in OC sources as well as in OCsed.

---

## Author Comment (AC1) · 7 Apr 2018

General comments

Comment #1: The paper was well executed and written and presented novel data on seagrass carbon dynamics. Particularly, this paper fills in a much needed gap on tropical blue carbon ecosystems and the contribution of belowground biomass (esp. sheathes) to carbon stocks, the latter often erroneously overlook or lumped in as the sediment carbon stock. It would be interesting to expand on this study by looking at

similar variables at deeper depths so that (a) it is comparable to global studies that look at 30-100 cm depths, and (b) we can understand better the long-term contributions of seagrass and allochthonous OC were living biomass isn't present and detritus has been processed more by microbial remineralization. There are some concerns about the lumping of different vegetation types into a site average, but otherwise these are minor revisions.

Reply #1: Thank you for your helpful comments. Please see our Reply #12 to your main concern.

Specific comments

Abstract

Comment #2: Lines 3, 11, 13: What is meant by enrichment? Looking at the next sentences, 'accumulation' may be a more accurate term. Change throughout the manuscript.

Reply #2: Concur.

Change #2: We have changed the term as per your suggestion.

Comment #3: Line 5: 'bodies' is an uncommon term for seagrasses and should be 'plants' or 'biomass' here and throughout the manuscript.

Reply #3: Concur.

Change #3: We have deleted 'bodies' or changed it to 'plants' or 'biomass' as per your suggestion.

Comment #4: It will be helpful to describe what species of seagrass are being studied in the abstract.

Reply #4: Concur.

Change #4: We have added "Thalassia hemprichii dominated" before "back-reef" and

"Enhalus acoroides dominated" before "estuarine sites" (page 2, line 7).

Comment #5: Line 16: no need to hyphenate blue carbon. Change throughout the manuscript as well.

Reply #5: Concur.

Change #5: We have removed the hyphen as per your suggestion.

Introduction

Comment #6: Lines 17-30: Consider Trevathan-Tackett et al. 2017 as a specific review of seagrass recalcitrance and the potential for contributing to OC stocks (doi: 10.3389/fpls.2017.00925); it will also be useful in the first section of the discussion. Also consider new research on providence of OC in seagrass meadows using eDNA: Reef et al 2017 doi: 10.1002/lno.10499

Reply #6: Concur with Trevathan-Tackett et al. 2017. However, we did not cite Reef et al. 2017 because our focus is not on the detailed provenance of OC but on factors controlling OC.

Change #6: We have added "; Trevathan-Tackett et al., 2017" to page 3 line 17.

Methods

Comment #7: How are you considering leaf detritus in these sediment measurements/ calculations? In sections 2.2 it says it's a part of the dead plant structures but not in the calculations. Is it assumed that 100% of the surface leaf detritus is exported and not buried?

Reply #7: Leaf detritus is included in the OC mass calculation (page 7 line 23 and lines 27–30) but not in the calculation of $\delta13C_{sed}$ (page 8 line 3). We have added the reason for its exclusion from the latter after the explanation of the calculation of $\delta13C_{sed}$.

Change #7: We have added the following sentences (page 8 line 4): "We did not in-

clude leaf detritus in the calculation of $\delta$13Csed because (1) the leaf fragments were so small that we could not remove epiphytes from them, and (2) their mass was much smaller than that of the sheath and rhizomes and roots, so we considered its contribution to $\delta$13Csed to be negligible."

Comment #8: Why is the Cfsed calculation multiplied by 1/3 (eqs. 6 & 8)?

Reply #8: We have multiplied by 1/3 because OCfsed is the averaged OC mass of the three layers (surface, medium, and bottom) of fine sediment.

Change #8: We have added the following sentence (page 7 line 30): "OCfsed is the averaged OC mass of the three layers (surface, medium, and bottom) of fine sediment".

Comment #9: How do equations 7 and 8 relation to traditional mixing model methods to look at OC providence? Were the end-members (seagrass, POM, algae/coral, terrestrial) taken into account? It seems a waste not to use this stable isotope to quantitatively obtain OC contribution values.

Reply #9: We intentionally did not use the stable isotope mixing model because, in the case examined in the present study, it failed to reliably isolate the contribution of seagrass from those of algae and corals; rather, the strong negative correlations among the inferred values imply that one source is simply being traded off against the other. (see Parnell et al., 2010). We showed that the direct supply of belowground seagrass detritus was a major mechanism of OCsed accumulation at the back-reef site from the contribution of belowground detritus to OCdead and $\delta$13Csed, and from the relationships among $\delta$13Csed, biomass, OCsed and OCdead (pages 11 lines 23–30).

Reference

Parnell, A. C., Inger, R., Bearhop, S., & Jackson, A. L.: Source partitioning using stable isotopes: coping with too much variation, PLOS ONE, 5, e9672, 2010, doi: 10.1371/journal.pone.0009672.

Results

Comment #10: Since section 3.2 only has one sentence, I'd suggest adding it to the next OC section.

Reply #10: Concur.

Change #10: We have added the sentence in section 3.2 to the next section and renumbered all sections in the Results.

Comment #11: One suggestion is to make a supplementary table(s) for the statistics. This would make reading the text easier.

Reply #11: We do not concur. We have left the statistics in the main text for the convenience of readers who wish to use the statistics to help them understand the results.

Comment #12: Where are the data on the differences between vegetated, unvegetated and bare OC stocks and fractions? This will be very important in the interpretation of OCbio and OCdead. This will give better resolution into the differences within and between back-reef and estuary regions.

Reply #12: Concur.

Change #12: As per your suggestion, we have added a figure showing the differences in total OC stock and its components between vegetated and no-vegetation (unvegetated and bare area) points (Fig. AC1). At both sites, OCbio, OCdead, OCfsed, OCsed, and OCtotal were significantly higher at points with vegetation than at points without vegetation. At points with vegetation, OCbio, OCcsed and OCtotal were significantly higher at the estuarine site than at the back-reef site, whereas OCdead, OCfsed, OCsed were not different between the sites. Therefore, this revision further supports our conclusion described in the original manuscript (page 12 line 24). Figure AC1 replaces Figure 4 in the revised manuscript (page 27) and the figure caption (page 22 lines 12–14) as well as the relevant results (page 9 lines 2–11) and discussion (page 12 line 24) have been modified accordingly.

Comment #13: What about correlations between living AG:BG?

Reply #13: We concur that the relationship between living AG:BG should be added.

Change #13: We have added a description of the relationship in the manuscript (page 10 line 7): "We also found significant positive correlations between aboveground and belowground biomass ($F_{1,18}$ = 94.10, P < 0.001, $r^2$ = 0.84)". We added the following sentence after "(Fig. 7c)." (page 10 line 12): "We also found significant positive correlations between aboveground and belowground biomass ($F_{1,6}$ = 78.40, P < 0.001, $r^2$ = 0.93)".

Discussion

Comment #14: Page 10, Lines 17-19: NO, we cannot assume constant to 1-m depth. There are important processes that affect OC down core, most notably the reduction on living biomass with depth, change in bulk density and microbial remineralization, so there is absolutely no meaning to the OCbio to OCtotal estimate. Please remove this sentence and calculation and find another more robust way to compare the OCbio data to previous literature.

Reply #14: We deleted the sentence as per your suggestion. Instead, we compared OCbio and OCtotal in this study with the above + belowground seagrass biomass OC and sedimentary OC in the top 0.15-m-thick layer, respectively, reported in a previous study (Fourqurean et al., 2012) (Table AC1).

Change #14: We have deleted the sentence (page 10 lines 17–21): "If we assume... (Fourqurean et al., 2012)". Instead, we compared data of OCbio and OCtotal in the present study with Fourqurean et al. (2012)'s data in the top 0.15-m-thick layer. We have added a new table (Table AC1) and the following sentence: "The averaged OCbio was significantly higher in this study than that in the previous study by Fourqurean et al. (2012) (W = 1691, P = 0.006), whereas the averaged OCsed was significantly lower in this study than in the previous study at both vegetated and no-vegetation points

(vegetated, W = 6952, P < 0.001; no-vegetation, W = 225, P = 0.039) (Table AC1). Hence, the contribution of OCbio to OCtotal at our sites was higher than the global average". We also changed "the highest in globally compiled data" to "higher than in globally compiled data" in the abstract (page 2 line 8).

Comment #15: Second paragraph: Anoxic sediments that generally reduce decomposition rates also can lead to higher preservation of OC.

Reply #15: True, but we did not add a statement about this effect to the main text because we were addressing the differences in the characteristics of OC accumulation in sediment between aboveground and belowground seagrass detritus.

Figures

Comment #16: Figure 1 is low quality and fuzzy and thus hard to read Figure 3: please define the abbreviations in the caption.

Reply #16: Concur.

Change #16: We have replaced Figure 1 with Figure AC2. We have defined the abbreviations in the caption of Figure 3.
* * *
**Table AC1. Values of seagrass biomass organic carbon and sedimentary organic carbon mass in globally compiled data (Fourqurean *et al.*, 2012) and this study (mean ± SD, *n*).**

| | Vegetated | | No-vegetation | |
|---|---|---|---|---|
| | Seagrass biomass OC (gC m$^{-2}$) | Sedimentary OC (gC L$^{-1}$) | Seagrass biomass OC (gC m$^{-2}$) | Sedimentary OC (gC L$^{-1}$) |
| | mean ± SD (*n*) | mean ± SD (*n*) | mean ± SD (*n*) | mean ± SD (*n*) |
| Fourqurean *et al.*, 2012 | 251.4 ± 395.6 (251) | 12.32 ± 8.04 (410) | - | 8.08 ± 5.90 (43) |
| This study | 283.0 ± 200.8 (21) | 5.03 ± 1.32 (21) | - | 2.93 ± 0.73 (7) |

11

**Fig. 1.** Table AC1

[Figure]

Figure AC1 : OC mass ($OC_{bio}$, $OC_{dead}$, $OC_{csed}$, $OC_{fsed}$, $OC_{sed}$, and $OC_{total}$) at (a) no-vegetation (bare and unvegetated) points at the back-reef site, (b) vegetated points at the back-reef site, (c) no-vegetation points at the estuarine site, and (d) vegetated points at the estuarine site. Boxes show the 25% and 75% quantiles; horizontal bands inside the boxes are median values; whiskers show maximum and minimum values; and the open circles are outliers.

**Fig. 2.** Figure AC1

[Figure]

Figure AC2: (a) (b) Study site location on Ishigaki Island, Japan. Sampling points at the (c) back-reef and (d) estuarine sites. At the back-reef site, the
circle indicating the southernmost vegetated sampling point actually represents a cluster of six sampling points.

**Fig. 3.** Figure AC2

[Figure]

---

## Author Comment (AC2) · 7 Apr 2018

General comments

Comment #1: The study by Tanaya et al. reports findings in the context of blue-carbon science, specifically as a representation for the Indo-Pacific region. The authors demonstrated meticulous planning for the study and the manuscript is generally well-written. Our group is similarly involved in blue carbon studies and we draw some corollary between this study and our findings. In addition, we suggest some recommendations that may improve the authors' present and future outlook in this field. One of the highlights of this study is the argument on the contribution of biomass derived organic carbon (OC bio) to the organic carbon pool (OC total) as the highest globally (P2L8). The data is presented in percentages (i.e. 19% OC bio and 81% OC sed) rather than the actual organic carbon stocks. It may be apt to complement such comparisons with actual global stock values (in equivalent measures as grams C per meter squared or megagrams C per hectare).

Reply #1: Concur.

Change #1: We have revised the data presentation to include a comparison of OC mass in this study with that of a previous study (Fourqurean et al., 2012) as per your suggestion. We have removed the sentence (page 10 lines 17–21): "If we assume... (Fourqurean et al., 2012)". Instead, we have added a new table (Table AC1) and the following sentence: "The averaged OCbio was significantly higher in this study than that in the previous study by Fourqurean et al. (2012) (W = 1691, P = 0.006), whereas the averaged OCsed was significantly lower in this study than in the previous study at both vegetated and no-vegetation points (vegetation, W = 6952, P < 0.001; no-vegetation, W = 225, P = 0.039) (Table AC1). Hence, the contribution of OCbio to OCtotal at our sites was higher than the global average". We have revised the phrase in the abstract (page 2 line 8) by replacing "the highest in globally compiled data" with "higher than globally compiled data".

Comment #2: They then rounded off their study by stating below-ground biomass is a driver for sediment OC storage (P2L14-15). It may hold true for this specific study, which is represented by findings from two sites. The authors rightly indicated past studies (e.g. Kennedy et al., 2004; 2010 and Howard et al. 2017 – P11L14) showed no relationships between seagrass biomass and sediment OC stocks. This is consistent with our recent study as well (Rozaimi et al. 2017). However, our other studies suggest otherwise whereupon biomass is indeed important in driving sediment OC stocks (Serrano et al. 2016; Rozaimi et al. 2013). Tanaya et al. provided possible explana-
tions on why they had different outcomes (P11) but alternatively, it may be plausible that their study sites may simply have exceptional sediment OC storage characteristics compared to other Indo-pacific seagrass meadows.

Reply #2: Concur.

Change #2: We have added the following sentence (page 11 line 18): "although we could not exclude the possibility that our sites may have specific sedimentary OC storage characteristics different from those of other Indo-Pacific seagrass meadows". We added the relevant literature (Rozaimi et al. 2017) after "Howard et al., 2017" (page 11 line 14).

Comment #3: Further to the above, it has to be clearly noted this study reports findings from surficial sediments (up to 16 cm depth: P5L5). This depth is within the range of vertical rhizomal growths for Indo-pacific seagrass rhizomes (especially T. hemprichii). So clearly autochthonous inputs play an important role in retaining seagrass-derived OC within this depth layer. However, the context of the authors' findings within 15 cm sediment depths up-scaled to 1 m, on the assumption that sediment OC density is constant (P10L18) may be too broad an assumption. In our published results (Rozaimi et al. 2017), we found variability in surficial downcore OC content (up to 30 cm sediment depth, albeit as %OC) as well as changing _13C sediment signatures with increasing sediment depth. In other studies (Rozaimi et al. in preparation), we did not find consistency in downcore OC content or OC density in cores up to 1 m. Conventionally, the scaling-up approach is employed (and admittedly we have used scaling-up approaches to model sediment OC stocks up to 1 m) to contextualise findings relative to regional and global estimates as that in Fourqurean et al. (2012). The authors' assumption in this regard may be corroborated if other evidence can be presented to support the notion of past seagrass occurrences in their study site (re: Serrano et al. 2016; Belshe et al. 2017). Or simply, such investigations may be room for improvements in the authors' future work.

Reply #3: Concur.

Change #3: We have deleted the sentence as per your suggestion. Instead, we have added a new table (Table AC1), see Change #1.

Comment #4: On a final note, it is particularly interesting the authors have data (though not apparently analysed as yet) that can be used in stable isotope mixing models. Mixing models have been increasingly used to account for the contributions of seagrass derived-OC to bulk sediment organic pool and could thus offer alternative insights to the authors' findings. We do wonder how the authors' approach in this study hold up compared to approaches such as stable isotope analysis in R (SIAR; e.g Watanabe and Kuwae 2015; Rozaimi et al. 2017) or eDNA approaches (Reef et al 2017). The lack of reference to SIAR, at least, is somewhat peculiar since there are co-authors in this current study, who are familiar with SIAR (i.e. Watanabe and Kuwae 2015). Overall, we view this study as interesting and may well be citable in future blue carbon endeavours.

Reply #4: We intentionally did not use the stable isotope mixing model because, in the case examined in the present study, it failed to reliably isolate the contribution of seagrass from those of algae and corals; rather, the strong negative correlations among the inferred values imply that one source is simply being traded off against the other. (see Parnell et al., 2010). We showed that the direct supply of belowground seagrass detritus was a major mechanism of OCsed accumulation at the back-reef site from the contribution of belowground detritus to OCdead and $\delta$13Csed, and from the relationships among $\delta$13Csed, biomass, OCsed and OCdead (pages 11 lines 23–30).

Reference Parnell, A. C., Inger, R., Bearhop, S., & Jackson, A. L.: Source partitioning using stable isotopes: coping with too much variation, PLOS ONE, 5, e9672, 2010, doi: 10.1371/journal.pone.0009672.

General technical comments:

Comment #5: Seagrass "bodies" is a peculiar term to use

Reply #5: Concur.

Change #5: We have deleted 'bodies' or replaced it with 'plants' or 'biomass'.

Comment #6: On the use of "enrichment": conventionally, communications in this regards may construe the presence of higher quantity of 13-C atoms (i.e. enriched samples) relative to non-enriched samples. In the text, readers may find some confusion on whether the authors refer to 13-C enrichment, or simply linguistic reference to higher amounts of a particular entity.

Reply #6: Concur.

Change #6: We have changed the term as per your suggestion.

Comment #7: P4L14-22: Content more suited in the Introduction section

Reply #7: We do not agree. We did not move these sentences because they are too long and detailed to be included in the introduction.

Comment #8: P11L14: A word missing after OC (perhaps OC stocks?)

Reply #8: Concur.

Change #8: We have modified "OC" to "%OC or OCmass" (page 11 line 14) as per your suggestion.

Comment #9: P21 Table 2: On data entries as 0.00 _ 0.00: do these data refer to nil values, or data values less than 0.001?

Reply #9: The data entries of 0.00 are values less than 0.01 g cm–3.

Change #9: We have changed the units of dry density from "g cm–3" to "mg cm–3" to avoid having entries of 0.00 (Table AC2).

Comment #10: P28 Figure 5: Axis labels are too small

[Figure]

Reply #10: Concur.

Change #10: We have enlarged axis labels of Figure 5 (Figure AC3).

**Table AC1. Values of seagrass biomass organic carbon and sedimentary organic carbon mass in globally compiled data (Fourqurean *et al.*, 2012) and this study (mean ± SD, *n*).**

| | Vegetated | | No-vegetation | |
| --- | --- | --- | --- | --- |
| | Seagrass biomass OC (gC m$^{-2}$) | Sedimentary OC (gC L$^{-1}$) | Seagrass biomass OC (gC m$^{-2}$) | Sedimentary OC (gC L$^{-1}$) |
| | mean ± SD (*n*) | mean ± SD (*n*) | mean ± SD (*n*) | mean ± SD (*n*) |
| Fourqurean *et al.*, 2012 | 251.4 ± 395.6 (251) | 12.32 ± 8.04 (410) | - | 8.08 ± 5.90 (43) |
| This study | 283.0 ± 200.8 (21) | 5.03 ± 1.32 (21) | - | 2.93 ± 0.73 (7) |

11

**Fig. 1.** Table AC1

**Table AC2: Organic carbon content, δ¹³C, and dry density of each sediment and dead plant component at the back-reef and estuarine sites.**

| | Back reef | | | Estuary | | |
|---|---|---|---|---|---|---|
| | Organic carbon | | Dry density (mg cm⁻³) mean ± SD (n) | Organic carbon | | Dry density (mg cm⁻³) mean ± SD (n) |
| | %OC (% DW) mean ± SD (n) | δ¹³C (‰ vs. VPDB) mean ± SD (n) | | %OC (% DW) mean ± SD (n) | δ¹³C (‰ vs. VPDB) mean ± SD (n) | |
| Fine sediment | 0.37 ± 0.13 (60) | −12.8 ± 0.8 (60) | 893 ± 303 (60) | 0.42 ± 0.20 (24) | −17.4 ± 3.6 (24) | 760 ± 294 (24) |
| Coarse sediment | 0.32 ± 0.13 (20) | −12.8 ± 1.1 (20) | 292 ± 152 (20) | 0.26 ± 0.08 (8) | −15.9 ± 1.5 (8) | 475 ± 142 (8) |
| Dead leaf | 24.80 ± 3.07 (3) | −8.9 ± 0.6 [a] (5) | 0.05 ± 0.04 (20) | 23.31 ± 3.86 [b] (3) | −9.3 ± 0.2 [a] (5) | 0.03 ± 0.04 (8) |
| Dead sheath and rhizome | 21.29 ± 4.07 (3) | −8.9 ± 0.6 [a] (5) | 0.55 ± 0.63 (20) | 27.52 ± 1.75 [b] (3) | −9.3 ± 0.2 [a] (5) | 1.44 ± 1.86 (8) |
| Dead root | 19.25 ± 1.67 (3) | −8.9 ± 0.6 [a] (5) | 0.26 ± 0.25 (20) | 19.94 ± 5.89 [b] (3) | −9.3 ± 0.2 [a] (5) | 0.31 ± 0.35 (8) |

[a]**Total of sheath and rhizomes, and root.**
[b]**At one sampling point (FS1) where the dominant species was different, the values were dead leaf, 25.77%; dead sheath and rhizome, 19.05%; and dead root, 19.21%.**

**Fig. 2.** Table AC2

[Figure]

Figure AC3: (a) $OC_{bio}$ (sum of aboveground and belowground biomass) (g C m$^{-2}$); (b) contribution of belowground biomass to $OC_{bio}$ (%); (c) $OC_{dead}$ (sum of above- and belowground detritus (g C m$^{-2}$); and (d) contribution of belowground detritus to $OC_{dead}$ (%). Boxes show the 25% and 75% quantiles; horizontal bands inside the boxes are median values; whiskers show maximum and minimum values; and open circles show outliers. (a) and (b) show the data of vegetated sampling points and (c) and (d) show the data of vegetated and bare sampling points.

**Fig. 3.** Figure AC3

---

## Author Comment (AC3) · 7 Apr 2018

Comment #1: This study aims to assess the mechanisms constraining organic carbon storage at two sites in Japan colonised by seagrass meadows quantifying the different pools of organic carbon that contribute to sediment organic carbon stock in seagrass sediments (and unvegetated sediments). The study demonstrates that seagrass structure and detritus constrain sediment organic carbon stores at the study sites. The manuscript is well written. However, I have some comments that I list in detail below.

Introduction. Page 3, line 33/Page 4, line 1. It is not clear in this sentence if the authors mean organic carbon or carbonate of calcareous organisms.

Reply #1: We have already clearly explained the meaning in the original manuscript: "OC derived from calcareous organisms" (page 3 line 33 and page 4 line 1).

Comment #2: Introduction. I suggest to re-write the last paragraph of the introduction to highlight the novel aspects of the study.

Reply #2: Concur.

Change #2: We have revised the phrase "the relationship between seagrass and the sedimentary OC stock" to "the pathways of sedimentary OC accumulation in seagrass meadows, especially the direct supply of belowground seagrass detritus" (page 4 line 7), and we added "along a seagrass biomass gradient" at the end of the paragraph (page 4 line 9).

Comment #3: Methods. Study site. The first paragraph could be moved to the introduction.

Reply #3: We do not agree. We did not move the paragraph because the description is too long and detailed to be included in the introduction.

Comment #4: Methods, page 4 last paragraph and Fig. 1. The location of the river mouth of Todoroki River relative to the sampling site is not clearly shown in the figure. This prevents to understand why the terrestrial input in this site is low. Similarly, the location of the small river discharging into the estuary is not clear in the image.

Reply #4: Concur.

Change #4: We have replaced Figure 1 with Figure AC2.

Comment #5: Methods. Page 5. It is not clear the type of organic material included in the fraction OCcsed. If it contained the carbonate from skeletons of corals, foraminifera, and other calcareous organisms it should not be considered in the organic carbon pool.

[Figure]

Reply #5: We have already explained that the carbonate was not included in OCcsed in the original manuscript (page 6 line 24).

Change #5: We have added "OC in the" before "coarse (> 1 mm diameter) sediments" (page 5 line 18) for clarity.

Comment #6: Methods. Page 5. Line 24. "We merged dead plant structures attached to live seagrass bodies into OCbio". How much did dead plant structures attached to living biomass weight? How much was it in comparison to mass of the seagrass dead compartment? Could this affect the OC results across compartments?

Reply #6: We have already explained in the original manuscript that their mass was usually very small (page 5 line 25).

Comment #7: Methods. Page 5, last paragraph. At each site, samples were collected in vegetated, unvegetated patches within the meadows and bare sediment. However the results in the box plots (Figs. 4 and 5) are presented per site, without indicating if they correspond to vegetated, unvegetated patches or bare sediment. I think it would be relevant to present these results indicating if the sediments were vegetated or not.

Reply #7: Concur.

Change #7: As per your suggestion, we have added a figure showing the differences in total OC stock and its components between vegetated and no-vegetation (unvegetated and bare area) points (Fig. AC1). At both sites, OCbio, OCdead, OCfsed, OCsed, and OCtotal were significantly higher at points with vegetation than at points without vegetation. At points with vegetation, OCbio, OCcsed and OCtotal were significantly higher at the estuarine site than at the back-reef site, whereas OCdead, OCfsed, OCsed were not different between the sites. Therefore, this revision further supports our conclusion described in the original manuscript (page 12 line 24). Figure AC1 replaces Figure 4 in the revised manuscript (page 27) and the figure caption (page 22 lines 12–14) as well as the relevant results (page 9 lines 2–11) and discussion (page 12 line 24) have been

modified accordingly. We have added an explanation of vegetation in the caption of Figure 5: Figure 5 (a) and (b) show the data of vegetated sampling points and Figure 5 (c) and (d) show the data of vegetated and bare sampling points.

Comment #8: Table 2. In this table the density of dead plant material is 0.00 _ 0.00 g cm-3. I believe that these components did have some dry density but lower than 0.00 g cm-3. I order to be able to provide their dry density, the units could be expressed in mg cm-3.

Reply #8: Concur.

Change #8: We have changed the units of dry density from "g cm–3" to "mg cm–3" to avoid entries of 0.00 (Table AC2).

Comment #9: Discussion. How much was the OC sediment stock at the studied seagrass meadows and at the bare sites? How do the OC stocks in the seagrass sediments found in this study compare with global seagrass OC sed stocks?

Reply #9: Concur. We have added these results and a corresponding explanation.

Change #9: We have removed the sentence (page 10 lines 17–21): "If we assume... (Fourqurean et al., 2012)". Instead, we compared data of OCbio and OCtotal in the present study with Fourqurean et al. (2012)'s data in the top 0.15-m-thick layer. We have added a new table (Table AC1) and the following sentence: "The averaged OCbio was significantly higher in this study than that in the previous study by Fourqurean et al. (2012) (W = 1691, P = 0.006), whereas the averaged OCsed was significantly lower in this study than in the previous study at both vegetated and no-vegetation points (vegetation, W = 6952, P < 0.001; no-vegetation, W = 225, P = 0.039) (Table AC1). Hence, the contribution of OCbio to OCtotal at our sites was higher than the global average". We also changed "the highest in globally compiled data" to "higher than in globally compiled data" in the abstract (page 2 line 8).

Comment #10: Discussion. What is the contribution of the different potential OC

sources (seagrass, algae, corals, suspended POM and terrestrial POM) to OC in the sediment at both sites (and discriminating between vegetated and bare sediment)? The fraction of the different sources to the compartments of coarse and fine sediment could be estimated using mixing models. These estimates could be incorporated in a revised Fig. 8.

Reply #10: We intentionally did not use the stable isotope mixing model because, in the case examined in the present study, it failed to reliably isolate the contribution of seagrass from those of algae and corals; rather, the strong negative correlations among the inferred values imply that one source is simply being traded off against the other. (see Parnell et al., 2010). We showed that the direct supply of belowground seagrass detritus was a major mechanism of OCsed accumulation at the back-reef site from the contribution of belowground detritus to OCdead and $\delta$13Csed, and from the relationships among $\delta$13Csed, biomass, OCsed and OCdead (pages 11 lines 23–30).

Reference

Parnell, A. C., Inger, R., Bearhop, S., & Jackson, A. L.: Source partitioning using stable isotopes: coping with too much variation, PLOS ONE, 5, e9672, 2010, doi: 10.1371/journal.pone.0009672.

Comment #11: Conclusions. Kennedy et al 2010 and several other papers demonstrate that the contribution of particle trapping and seagrass material to sediment organic carbon widely varies across seagrass meadows, from meadows where allochthonous carbon is the main source to others where the sediment organic carbon pool is dominated by seagrass material. Therefore, there is evidence in the literature that seagrass carbon can be an important source to sediment organic carbon.

Reply #11: Although previous studies showed the provenance of sedimentary OC, they did not show the pathway of sedimentary OC (page 3 lines 28–30). We empirically showed that not only suspended-particle trapping but also the direct supply of belowground seagrass detritus can be a dominant organic carbon accumulation pathway in

seagrass sediments.

Minor comments

Comment #12: Minor comments Abstract- line 7. It should say that the stable carbon isotope ratio was measured in OC sources as well as in OCsed.

Reply #12: Concur.

Change #12: We have added "and its potential OC sources" after "($\delta$13C) of OCsed" (page2 line 7).
* * *
**Table AC1. Values of seagrass biomass organic carbon and sedimentary organic carbon mass in globally compiled data (Fourqurean *et al.*, 2012) and this study (mean ± SD, *n*).**

| | Vegetated | | No-vegetation | |
|---|---|---|---|---|
| | Seagrass biomass OC (gC m$^{-2}$) | Sedimentary OC (gC L$^{-1}$) | Seagrass biomass OC (gC m$^{-2}$) | Sedimentary OC (gC L$^{-1}$) |
| | mean ± SD (*n*) | mean ± SD (*n*) | mean ± SD (*n*) | mean ± SD (*n*) |
| Fourqurean *et al.*, 2012 | 251.4 ± 395.6 (251) | 12.32 ± 8.04 (410) | - | 8.08 ± 5.90 (43) |
| This study | 283.0 ± 200.8 (21) | 5.03 ± 1.32 (21) | - | 2.93 ± 0.73 (7) |

11

**Fig. 1.** Table AC1

**Table AC2: Organic carbon content, δ¹³C, and dry density of each sediment and dead plant component at the back-reef and estuarine sites.**

| | Back reef | | | Estuary | | |
|---|---|---|---|---|---|---|
| | Organic carbon | | Dry density (mg cm$^{-3}$) mean ± SD ($n$) | Organic carbon | | Dry density (mg cm$^{-3}$) mean ± SD ($n$) |
| | %OC (% DW) mean ± SD ($n$) | $\delta^{13}$C (‰ vs. VPDB) mean ± SD ($n$) | | %OC (% DW) mean ± SD ($n$) | $\delta^{13}$C (‰ vs. VPDB) mean ± SD ($n$) | |
| Fine sediment | 0.37 ± 0.13 (60) | −12.8 ± 0.8 (60) | 893 ± 303 (60) | 0.42 ± 0.20 (24) | −17.4 ± 3.6 (24) | 760 ± 294 (24) |
| Coarse sediment | 0.32 ± 0.13 (20) | −12.8 ± 1.1 (20) | 292 ± 152 (20) | 0.26 ± 0.08 (8) | −15.9 ± 1.5 (8) | 475 ± 142 (8) |
| Dead leaf | 24.80 ± 3.07 (3) | −8.9 ± 0.6 [a] (5) | 0.05 ± 0.04 (20) | 23.31 ± 3.86 [b] (3) | −9.3 ± 0.2 [a] (5) | 0.03 ± 0.04 (8) |
| Dead sheath and rhizome | 21.29 ± 4.07 (3) | −8.9 ± 0.6 [a] (5) | 0.55 ± 0.63 (20) | 27.52 ± 1.75 [b] (3) | −9.3 ± 0.2 [a] (5) | 1.44 ± 1.86 (8) |
| Dead root | 19.25 ± 1.67 (3) | −8.9 ± 0.6 [a] (5) | 0.26 ± 0.25 (20) | 19.94 ± 5.89 [b] (3) | −9.3 ± 0.2 [a] (5) | 0.31 ± 0.35 (8) |

[a] Total of sheath and rhizomes, and root.
[b] At one sampling point (FS1) where the dominant species was different, the values were dead leaf, 25.77%; dead sheath and rhizome, 19.05%; and dead root, 19.21%.

**Fig. 2.** Table AC2

[Figure]

Figure AC1 : OC mass (OC_bio, OC_dead, OC_csed, OC_fsed, OC_sed, and OC_total) at (a) no-vegetation (bare and unvegetated) points at the back-reef site, (b) vegetated points at the back-reef site, (c) no-vegetation points at the estuarine site, and (d) vegetated points at the estuarine site. Boxes show the 25% and 75% quantiles; horizontal bands inside the boxes are median values; whiskers show maximum and minimum values; and the open circles are outliers.

**Fig. 3.** Figure AC1

[Figure]

Figure AC2: (a) (b) Study site location on Ishigaki Island, Japan. Sampling points at the (c) back-reef and (d) estuarine sites. At the back-reef site, the circle indicating the southernmost vegetated sampling point actually represents a cluster of six sampling points.

**Fig. 4.** Figure AC2